# Physical basis of the cell size scaling laws

Romain Rollin[1]*, Jean-François Joanny[1,2]*, Pierre Sens[1]*

[1]Institut Curie, PSL Research University, CNRS UMR168, Paris, France; [2]Collège de France, Paris, France

**Abstract** Cellular growth is the result of passive physical constraints and active biological processes. Their interplay leads to the appearance of robust and ubiquitous scaling laws relating linearly cell size, dry mass, and nuclear size. Despite accumulating experimental evidence, their origin is still unclear. Here, we show that these laws can be explained quantitatively by a single model of size regulation based on three simple, yet generic, physical constraints defining altogether the Pump-Leak model. Based on quantitative estimates, we clearly map the Pump-Leak model coarse-grained parameters with the dominant cellular components. We propose that dry mass density homeostasis arises from the scaling between proteins and small osmolytes, mainly amino acids and ions. Our model predicts this scaling to naturally fail, both at senescence when DNA and RNAs are saturated by RNA polymerases and ribosomes, respectively, and at mitotic entry due to the counterion release following histone tail modifications. Based on the same physical laws, we further show that nuclear scaling results from a osmotic balance at the nuclear envelope and a large pool of metabolites, which dilutes chromatin counterions that do not scale during growth.

## Editor's evaluation

This important theoretical work deals with the problem of homeostasis of protein density within cells, relying on the Pump and Leak model. The model makes predictions both for growing and senescent cells, which they convincingly compare to experimental data on budding yeast. The authors further study the long-standing problem of nuclear scaling (the constant ratio between the nucleus and cell volume), and show that within their model it arises naturally from osmotic balance at the nuclear envelope, with metabolites playing a major role.

*For correspondence:
romain.rollin@curie.fr (RR);
jean-francois.joanny@college-de-france.fr (J-FJ);
pierre.sens@curie.fr (PS)

## Introduction

Although cell size varies dramatically between cell types, during the cell cycle, and depends on various external stresses (*Cadart et al., 2019*), each cell type often shows small volumetric variance at a given stage of its cycle. This tight control reflects the importance of size in monitoring cell function. It is often associated to generic linear scaling relations between cell volume, cell dry mass, and the volume of the nucleus (*Neurohr and Amon, 2020*; *Cantwell and Nurse, 2019b*; *Webster et al., 2009*). These scaling laws have fascinated biologists for more than a century (*Wilson, 1925*; *Conklin, 1912*) because of the inherent biological complexity and their ubiquity both in yeasts, bacteria, and mammals, hence raising the question of the underlying physical laws.

Although robust, these scaling relations do break down in a host of pathologies. The nuclear-to-cytoplasmic (NC) ratio (also called karyoplasmic ratio) has long been used by pathologists to diagnose and stage cancers (*Jevtić and Levy, 2014*; *Slater et al., 2005*; *Zink et al., 2004*). Similarly, senescent cells such as fibroblasts are known to be swollen and their dry mass diluted (*Neurohr et al., 2019*), a feature suspected to be of fundamental biological importance since it could represent a determinant of stem cell potential during ageing (*Lengefeld et al., 2021*).

Paradoxically, although cells are the simplest building blocks of living organisms, the principles ruling their size, their growth, and the associated scaling laws have not yet been fully resolved. This

is in part due to the experimental difficulty to perform accurate volume and dry mass measurements (*Model, 2018*; *Cadart et al., 2017*; *Park et al., 2018*). Many methods were developed in the past decades, but they sometimes lead to contradictory observations, highlighting the need of comparing and benchmarking each method (*Guo et al., 2017*; *Venkova et al., 2022*).

Moreover, extensive experimental investigations have identified a plethora of biological features influencing these scalings (*Lang et al., 1998*; *Hoffmann et al., 2009*) but comparatively fewer theoretical studies have precisely addressed them, leaving many experimental data unrelated and unexplained. Several phenomenological theories have emerged to understand individual observations, but they are still debated among biologists. The 'nucleoskeletal theory' emphasizes the role of the DNA content in controlling the NC ratio based on the idea that ploidy dictates cell and nuclear sizes since tetraploid cells tend to be larger than their diploid homologs (*Webster et al., 2009*). Other experiments suggest that genome size is not the only determining factor: indeed it would not explain why cells from different tissues, having the same amount of DNA, have different sizes. Instead, it has been shown that nuclear size depends on cytoplasmic content, nucleo-cytoplasmic transport, transcription, RNA processing, and mechanics of nuclear envelope structural elements such as lamina (*Cantwell and Nurse, 2019b*).

In parallel, theoretical models, based on nonequilibrium thermodynamics, were developed (*Kedem and Katchalsky, 1958*; *Mori, 2012*; *Marbach and Bocquet, 2019*), often based on the 'Pump-Leak' principle (*Cadart et al., 2019*; *Venkova et al., 2022*; *Adar and Safran, 2020*). Charged impermeant molecules in cells create an imbalance of osmotic pressure at the origin of the so-called Donnan effect (*Sten-Knudsen, 2007*). Cells have two main ways to counteract the osmotic imbalance. They can adapt to sustain a high hydrostatic pressure difference as plants do by building cellulose walls. Or, as done by mammalian cells, they can use ion pumps to actively decrease the number of ions inside the cells, thus decreasing the osmotic pressure difference across the cell membrane and therefore impeding water penetration. However, due to the large number of parameters of these models, we still have a poor understanding of the correspondence between biological factors and physical parameters of the model.

In this article, we bridge the gap between phenomenological and physical approaches by building a minimal framework based on a nested Pump-Leak model to understand the cell size scaling laws as well as their breakdown in specific cases. Performing order of magnitude estimates, we precisely map the coarse-grained parameter of a simplified version of the Pump-Leak model to the main microscopic biological components. We find that the dry mass of the cell is dominated by the contribution of the proteins, while the cell volume is mostly fixed by the contribution to the osmotic pressure of small osmolytes, such as amino acids and ions. The maintenance of a homeostatic cell density during growth is then due to a linear scaling relation between protein and small osmolyte numbers. Combining simplified models of gene transcription and translation and of amino acid biosynthesis to the Pump-Leak model, we show that the linear scaling relation between protein and small osmolyte numbers is obtained in the exponential growth regime of the cell by virtue of the enzymatic control of amino acid production.

On the other hand, the absence of linear scaling relation between protein and small osmolyte numbers is at the root of the breakdown of density homeostasis. This conclusion is in line with two biological mechanisms that were proposed to explain the regulation of size and density respectively during cell spreading and under hyper-osmotic shocks. Recent studies *Venkova et al., 2022*; *Adar and Safran, 2020* have indeed shown that the change of mechanical tension at the plasma membrane during cell spreading alters the permeability of ion channels and results in a volume adaptation at constant dry mass. Another important way through which cells control their size independently of their mass is through metabolite synthesis. An example is the synthesis of glycerol that occurs in budding yeast following a hypertonic shock (*Neurohr and Amon, 2020*). While these two mechanisms are now well established, here, we propose a new physical interpretation of two other important biological events, namely the dilution that occurs at senescence and the dilution that occurs at the beginning of mitosis, due to two distinct physical phenomena. At senescence, cells cannot divide properly. Our theory then predicts that DNA and RNAs become saturated by RNA polymerases (RNAPs) and ribosomes, respectively, leading to a change of the growth regime: the protein number saturates while the amino acid number increases linearly with time, resulting in the experimentally observed dry mass dilution. This prediction is quantitatively tested using published data of growing yeast cells prevented

from dividing (*Neurohr et al., 2019*). At mitotic entry, chromatin rearrangements, such as histone tail modifications, induce a counterion release inside the cell, resulting in an influx of water and dry mass dilution in order to maintain the osmotic balance at the cell membrane.

Finally, to further illustrate the generality of our model, we show that the linear scaling of nucleus size with cell size originates from the same physical effects. Using a nested Pump-Leak model for the cell and for the nucleus, we show that nuclear scaling requires osmotic balance at the nuclear envelope. The osmotic balance is explained by the nonlinear osmotic response of mammalian nuclei that we attribute to the presence of folds at the surface of many nuclei (*Lomakin et al., 2020*), which in turn buffer the nuclear envelope tension and enforce scaling. Nonetheless, the condition on osmotic balance appears to be insufficient to explain the robustness of the NC ratio during growth. Counterintuitively, metabolites, though permeable to the nuclear envelope, are predicted to play an essential role in the NC ratio. Their high concentrations in cells, a conserved feature throughout living cells, is shown to dilute the chromatin counterions that do not scale during growth, thereby allowing the scaling of nuclear size with cell size both at the population level and during individual cell growth.

## Results

### Pump-Leak model

Our theoretical approach to understand the various scaling laws associated to cell size is based on the Pump-Leak model (*Tosteson and Hoffman, 1960* and *Figure 1A*). The Pump-Leak model is a coarse-grained model emphasizing the role of mechanical and osmotic balance. The osmotic balance involves two types of osmolytes, impermeant molecules such as proteins and metabolites, which cannot diffuse through the cell membrane, and ions, which cross the cell membrane and for which at steady state the incoming flux into the cell must equate the outgoing flux. For simplicity, we restrict ourselves to a two-ion Pump-Leak model where only cations are pumped outward of the cell. We justify in the 'Discussion', section 'Physical grounds of the model', why this minimal choice is appropriate for the purpose of this article. Within this framework, three fundamental equations determine the cell volume. (1) Electroneutrality: the laws of electrostatics ensure that in any aqueous solution such as the cytoplasm, the solution is neutral at length scales larger than the Debye screening length, that is, the electrostatic charge of any volume larger than the screening length vanishes. In physiological conditions, the screening length is typically on the nanometric scale. Therefore, the mean charge density of the cell vanishes in our coarse-grained description (*Equation A.1*). (2) Osmotic balance: balance of the chemical potential of water inside and outside the cell; the timescale to reach the equilibrium of water is of few seconds after a perturbation (*Venkova et al., 2022*; *Milo and Phillips, 2015*). Note that the second part of the equality in *Equation 2* assumes that both the cellular and outer media are dilute solutions. This assumption may seem odd since the cytoplasm is crowded. Yet, we justify this approximation in the 'Discussion' (see section 'Physical grounds of the model'). (3) Balance of ionic fluxes: the typical timescales of ion relaxation observed during a cell regulatory volume response after an osmotic shock are of the order of a few minutes (*Venkova et al., 2022*; *Hoffmann et al., 2009*). Together, this means that our quasi-static theory is designed to study cell volume variations on timescales larger than a few minutes. We emphasize that although *Equation 2* and *Equation 3* are stationary equations, the volume of the cell may vary, for example, through the biosynthesis of impermeant molecules $X$ (see *Equation 4*), but this is a slow process at the timescale of equilibration of water and ion fluxes and we use a quasistatic approximation. Mathematically, the three equations read (see Appendix 1, section 1.1 for the full derivations of these equations):

$$n^+ - n^- - z \cdot x = 0 \tag{1}$$

$$\Delta P = \Delta \Pi = k_B T \cdot \left( n^+ + n^- + x - 2 n_0 \right) \tag{2}$$

$$n^+ \cdot n^- = \alpha_0 \cdot n_0^2 \tag{3}$$

where, $n^+$, $n^-$ are respectively the ionic concentrations of positive and negative ions inside the cell. The concentrations of cations and anions are equal in the external medium to enforce electroneutrality since the concentrations of non-permeant molecules in the outer medium are typically much lower than their ionic counterparts (*Milo and Phillips, 2015*). We call $n_0$ the corresponding concentrations, and the associated osmotic pressure in the outer medium reads $\Pi_0 = k_B T \cdot 2 n_0$. The cell is modeled

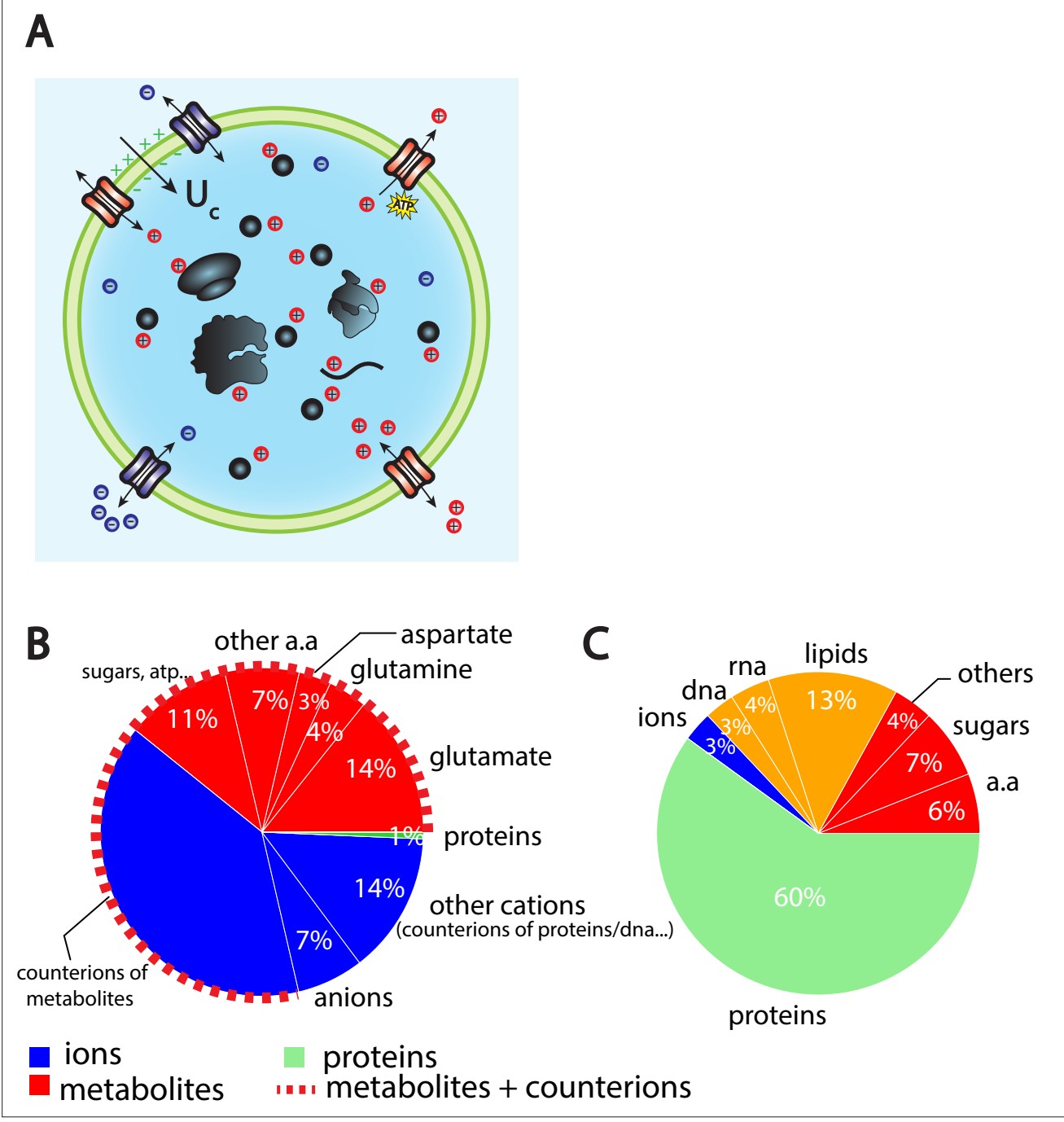

**Figure 1.** The Pump-Leak model coupled to order of magnitude estimates predicts that the cellular wet volume is mainly accounted for by metabolites (mainly amino acids, mostly glutamate) and their counterions in mammalian cells. (**A**) Schematic of the Pump-Leak model. Species in black are impermeant molecules such as proteins, mRNAs, and metabolites (black circles). In average, those molecules are negatively charged and thus carry positive counterions (red species) to ensure electroneutrality. Ions can freely cross the plasma membrane through channels. Their concentrations in the cell result from the balance of three fluxes: the electrical conduction, the entropic diffusion, and pumping. In the model, only cations are pumped out of the cell to model the Na/K pump but this assumption is not critical (see 'Discussion', section 'Physical grounds of the model' and Appendix 1, section 2.1.3). (**B**) Fraction of volume and (**C**) of the dry mass occupied by the constituents of a mammalian cell (see Appendix 1, section 3.1 and *Alberts et al., 2002*).

**Table 1.** Estimation of the coarse-grained Pump-Leak model parameters for a typical Mammalian cell.

| Symbol | Typical value | Meaning |
|---|---|---|
| $n^+$ | 160 mmol | Cation concentration (**Milo and Phillips, 2015**) |
| $n^-$ | 20 mmol | Anion concentration (**Milo and Phillips, 2015**) |
| $n_0$ | 150 mmol | External cationic/anionic concentrations (**Milo and Phillips, 2015**) |
| $\Delta P$ | 10–100 Pa | Difference of hydrostatic pressure through the plasma membrane (**Smeets et al., 2019**; **Pegoraro et al., 2017**) |
| $\frac{R}{V}$ | 30% | Volume fraction occupied by the dry mass (**Venkova et al., 2022**; **Alberts et al., 2002**) |
| $\alpha_0$ | 0.14 | Dimensionless parameter comparing pumping and passive leaking of cations (**Equation 3**) |
| $U_c$ | –52 mV | Cytoplasmic membrane potential (**Equation A.8**) |
| $\Pi_0$ | $7.2 \cdot 10^5$ Pa | External osmotic pressure |
| $\Pi$ | $7.2 \cdot 10^5$ Pa | Cellular Van't Hoff osmotic pressure (**Equation 2**) |
| $x$ | 120 mmol | Cellular concentrations of species other than ions (**Equation 2**) |
| $-z$ | –1.2 | Average charge of species other than ions (**Equation 1**) |

as a compartment of total volume $n_0$ divided between an excluded volume occupied by the dry mass $R$ and a wet volume. The cell contains ions and impermeant molecules such as proteins, RNA, free amino acids, and other metabolites. The number $X$, respectively the concentration $x = \frac{X}{V-R}$, of these impermeant molecules may vary with time due to several complex biochemical processes such as transcription, translation, plasma membrane transport, and degradation pathways. The average negative charge $-z$ of these trapped molecules induces a Donnan potential difference $U_c$ across the cell membrane. The Donnan equilibrium contributes to the creation of a positive difference of osmotic pressure $\Delta\Pi$ that inflates the cell. Cells have two main ways to counteract this inward water flux. They can either build a cortex stiff enough to prevent the associated volume increase, as done by plant cells. This results in the appearance of a hydrostatic pressure difference $\Delta P$ between the cell and the external medium. Or they can pump ions outside the cell to decrease the internal osmotic pressure, a strategy used by mammalian cells. We introduce a pumping flux of cations $p$. Cations can also passively diffuse through the plasma membrane via ion channels with a conductivity $g^+$. In **Equation 3**, the pumping efficiency is measured by the dimensionless number $\alpha_0 = e^{-\frac{p}{k_B T g^+}}$ where $T$ is the temperature and $k_B$ the Boltzmann constant. The pumping efficiency varies between 0 in the limit of 'infinite pumping' and 1 when no pumping occurs (see Appendix 1, section 1.1 for an explanation on the origin of this parameter).

## Volume and dry mass scaling

Although proposed more than 60 years ago (**Tosteson and Hoffman, 1960**) and studied in depth by mathematicians (**Mori, 2012**) and physicists (**Kay, 2017**), little effort has been done to precisely map the coarse-grained parameters of the Pump-Leak model to microscopic parameters. We adopt here the complementary strategy and calculate orders of magnitude in order to simplify the model as much as possible, only keeping the leading order terms. We summarize in **Table 1** the values of the Pump-Leak model parameters that we estimated for a 'typical' mammalian cell. Three main conclusions can

be drawn: (1) pumping is important, as indicated by the low value of the pumping efficiency $\alpha_0 \sim 0.14$. Analytical solutions presented in the main text will thus be given in the 'infinite pumping' limit, that is, $\alpha_0 \sim 0$, corresponding to the scenario where the only ions present in the cell are the counterions of the impermeant molecules (see Appendix 1, section 2.1 for the general solutions). Though not strictly correct, this approximation gives a reasonable error of the order 10% on the determination of the volume due to the typical small concentration of free anions in cells (*Table 1*). This error is comparable to the typical volumetric measurement errors found in the literature. (2) Osmotic pressure is balanced at the plasma membrane of a mammalian cell since hydrostatic and osmotic pressures differ by at least three orders of magnitude. This result implies that even though the pressure difference $\Delta P$ plays a significant role in shaping the cell, it plays a negligible role in fixing the volume (see *Equation A.15* for justification). (3) The cellular density of impermeant species is high, $x \sim 120\text{mMol}$, comparable with the external ionic density $n_0$.

In this limit, the volume of the cell hence reads (the complete expression is given in Appendix 1, section 2.1):

$$V = R + \frac{(z+1) \cdot X}{2n_0} \tag{4}$$

The wet volume of the cell is thus slaved to the number of impermeant molecules that the cell contains. While this conclusion is widely acknowledged, the question is to precisely decipher which molecules are accounted for by the number X. We first estimate the relative contributions of the cellular-free osmolytes to the volume of the cell and then compute their relative contributions to the dry mass of the cell. We provide a graphical summary of our orders of magnitudes in *Figure 1B and C* as well as the full detail of their derivations in Appendix 1, section 3.1. The conclusion is twofold. Metabolites and their counterions account for most of the wet volume of the cell, 78% of the wet volume against 1% for proteins. On the other hand, proteins account for most of the dry mass of mammalian cells, accounting for 60% of the cellular dry mass against 17% for metabolites.

We further note that metabolites are mainly amino acids and in particular three of them, glutamate, glutamine, and aspartate, account for 73% of the metabolites (*Park et al., 2016*). It is important to note that the relative proportion of free amino acids in the cell does not follow their relative proportion in the composition of proteins. For instance, glutamate represents 50% of the free amino acid pool while its relative appearance in proteins is only 6% (*King and Jukes, 1969*). This is evidence that some amino acids have other roles than building up proteins. In particular, we demonstrate throughout this article their crucial role on cell size and its related scaling laws.

These conclusions may appear surprising due to the broadly reported linear scaling between cell volume (metabolites) and dry mass (proteins), hence enforcing a constant dry mass density $\rho$ during growth. Theoretical papers often assume a linear phenomenological relation between volume and protein number in order to study cell size (*Lin and Amir, 2018*; *Wu et al., 2022*). Our results instead emphasize that the proportionality is indirect, only arising from the scaling between metabolites (mostly amino acids) and proteins. The dry mass density reads (to lowest order):

$$\rho = \frac{M}{V} \approx \frac{\mathcal{M}_a \cdot l_p \cdot P_{tot}}{v_p \cdot P_{tot} + \frac{(z_{A^f}+1) \cdot A^f}{2n_0}} \tag{5}$$

where $\mathcal{M}_a$, $z_{A^f}$, and $A^f$ are respectively the average mass, charge, and number of amino acids; $l_p$, $v_p$, and $P_{tot}$ the average length, excluded volume, and number of proteins. Note that density homeostasis is naturally achieved in the growth regime where $A^f$ is proportional to $P_{tot}$.

## Model of gene expression and translation

To further understand the link between amino acid and protein numbers, we build upon a recent model of gene expression and translation (*Lin and Amir, 2018* and *Figure 2A*). The key feature of this model is that it considers different regimes of mRNA production rate $\dot{M}_j$ and protein production rate $\dot{P}_j$ according to the state of saturation of respectively the DNA by RNA polymerases (RNAPs) and mRNAs by ribosomes. For the sake of readability, we call enzymes both ribosomes and RNAPs, their substrates are respectively mRNAs and DNA and their products proteins and mRNAs. The scenario of the model is the following. Initially, the majority of enzymes are bound to their substrates and occupy a small fraction of all possible substrate sites. In this nonsaturated regime, that is, when the number

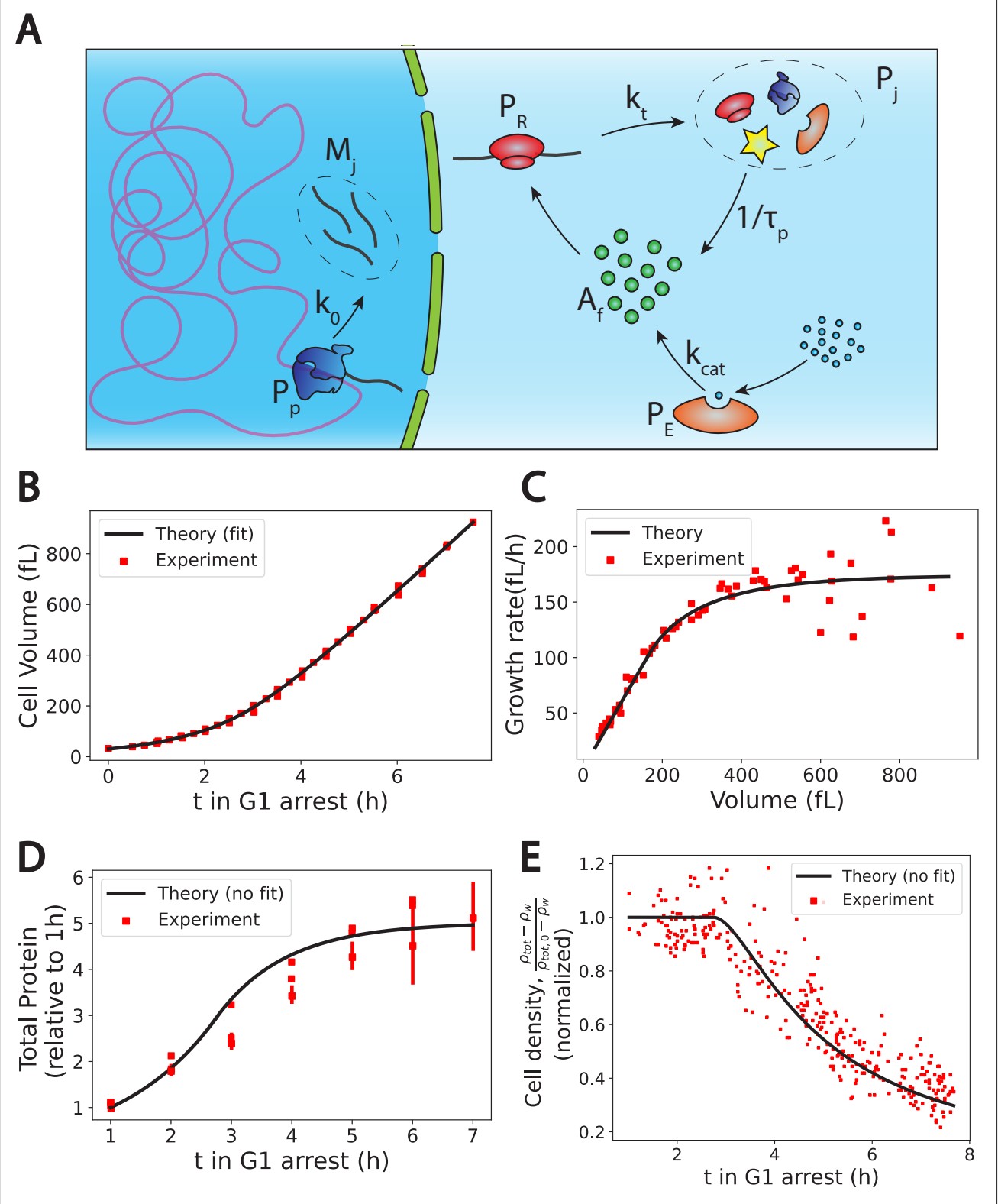

**Figure 2.** The Pump-Leak model coupled to a growth model predicts quantitatively dry mass homeostasis and its subsequent dilution at senescence. (**A**) Schematic of the growth model. RNAPs ($P_p$) transcribe DNA and form mRNAs ($M_j$) at an average rate $k_0$. mRNAs are then read by ribosomes ($P_r$) to produce proteins ($P_j$) at an average rate $k_t$. Proteins are degraded at an average rate $\frac{1}{\tau_p}$ into free amino-acids ($A_f$). Free amino acids are also synthetized from nutrients (blue circles) at a rate $k_{cat}$. This reaction is catalyzed by enzymes ($P_e$). (**B–E**) Comparison between theory (black) and

*Figure 2 continued on next page*

*Figure 2 continued*

experiment (red). (**B, C**, and **E**) have respectively been adapted from Figure 3.A, B and H of ***Neurohr et al., 2019***. (**D**) has been adapted from Figure 3.C,D and S5C of ***Neurohr et al., 2019***. (**D, E**) Model predictions without any fitting parameters. The total mass density $\rho_{tot}$ is defined as the total mass of the cell (water and dry mass) over the total volume of the cell. The online version of this article includes the following source data for ***Figure 2***: ***Figure 2—source data 1–4***. Data tables extracted from ***Neurohr et al., 2019***.

The online version of this article includes the following source data for figure 2:

**Source data 1.** Data tables extracted from Figure 3.A of ***Neurohr et al., 2019***.

**Source data 2.** Data tables extracted from Figure 3.B of ***Neurohr et al., 2019***.

**Source data 3.** Data tables extracted from Figure 3.C,D and S5C of ***Neurohr et al., 2019***.

**Source data 4.** Data tables adapted from Figure 3.H of ***Neurohr et al., 2019***.

of enzymes is smaller than a threshold value $P_p^*$ and $P_r^*$ (***Equation A.37***), the production rates of the products of type j read (***Lin and Amir, 2018***):

$$\dot{M}_j = k_0 \cdot \phi_j \cdot P_p - \frac{M_j}{\tau_m}, \qquad if \quad P_p \leq P_p^* \tag{6}$$

$$\dot{P}_j = k_t \cdot \frac{M_j}{\sum M_j} \cdot P_r - \frac{P_j}{\tau_p}, \qquad if \quad P_r \leq P_r^* \tag{7}$$

Both production rates have two contributions. (1) A source term characterized by the rates $k_0$ and $k_t$ at which the enzyme produces the product once it is bound to its substrate, times the average number of enzymes per substrate coding for the product of type j. This number is the fraction of substrates coding for product of type j – which can be identified as a probability of attachment ($\phi_j = \frac{g_j}{\sum g_j}$ and $\frac{M_j}{\sum M_j}$, where $g_j$, $M_j$ accounts for the number of genes and mRNAs coding for the product of type j) – multiplied by the total number of enzymes ($P_p$ and $P_r$). (2) A degradation term characterized by the average degradation times $\tau_m$ and $\tau_p$ of mRNAs and proteins. Note that we added a degradation term for proteins not present in ***Lin and Amir, 2018***, which turns out to be of fundamental importance below. Although these timescales vary significantly between species their ratio remains constant, $\tau_m$ being at least one order of magnitude smaller than $\tau_p$ in yeast, bacteria, and mammalian cells (***Milo and Phillips, 2015***). This justifies a quasistatic approximation, $\dot{M}_j \sim 0$, during growth such that the number of mRNAs of type j adjusts instantaneously to the number of RNAPs, in the nonsaturated regime:

$$M_j = k_0 \cdot \tau_m \cdot \phi_j \cdot P_p \tag{8}$$

During interphase, the number of enzymes grows, increasingly more enzymes attach to the substrates up to the saturation value due to their finite size. In this regime, we use the same functional form for the production rates only replacing the average number of enzymes per substrate by their saturating values: $g_j \cdot \mathcal{N}_p^{max}$ for RNAPs and $M_j \cdot \mathcal{N}_r^{max}$ for ribosomes (see Appendix 1, section 4.1 and ***Equations A.35 and A.36***), where $\mathcal{N}_p^{max}$ and $\mathcal{N}_r^{max}$ are the average maximal number of RNAPs and ribosomes per mRNAs and genes. Note that the model predicts that the saturation of DNA precedes that of mRNAs, whose number initially increases with the number of RNAPs (***Equation 8***) while the number of genes remains constant. We also highlight that a more general gene expression model was recently proposed (***Wang and Lin, 2021***), in which the saturation of DNA by RNAPs is due to a high free RNAP concentration near the promoter. Yet, we do not expect the exact saturation mechanism to change our conclusions. Once DNA is saturated, the number of mRNAs plateaus, leading to their saturation by ribosomes (see Appendix 1, section 4.1 and ***Equation A.41***).

Our previous analysis has highlighted the fundamental importance of free amino acids on cell volume regulation ***Figure 1B***. The production rate of free amino acids can be related to the number of enzymes catalyzing their biosynthesis using a linear process by assuming that the nutrients necessary for the synthesis are in excess:

$$\dot{A}^f = k_{cat} \cdot P_e - l_p \cdot \dot{P}_{tot} \tag{9}$$

where $k_{cat}$ is the rate of catalysis and $P_e$ the number of enzymes. The second term represents the consumption of amino acids to form proteins, with $P_{tot} = \sum P_j$. Although ***Equation 9*** is coarse-grained, we highlight that, since glutamate and glutamine are the most abundant amino acids in

the cell, it could in particular model the production of these specific amino acids from the Krebs cycle (**Alberts et al., 2002**). Note that we also ignored amino acid transport through the plasma membrane. The rationale behind this choice is twofold. (1) We do not expect any qualitative change when adding this pathway to our model since amino acid transport is also controlled by proteins. (2) We realized that the amino acids that actually play a role in controlling the volume, mainly glutamate, glutamine, and aspartate, are nonessential amino acids, hence that can be produced by the cell.

## Dry mass scaling and dilution during cell growth

We now combine the Pump-Leak model, the growth model, and the amino acid biosynthesis model to make predictions on the variation of the dry mass density during interphase. A crucial prediction of the growth model is that as long as mRNAs are not saturated, that is, $P_r < P_r^*$, all the protein numbers scale with the number of ribosomes, $P_j \sim \frac{\phi_j}{\phi_r} \cdot P_r$. Moreover, the autocatalytic nature of ribosome formation makes their number grow exponentially (**Equation 7**), that is, $P_r = P_{r,0} \cdot e^{k_r \cdot t}$, where $k_r = k_t \cdot \phi_r - \frac{1}{\tau_p}$ is the effective rate of ribosome formation (and also the rate of volume growth in this regime; **Equation A.39**) . The most important consequence of this exponential growth coupled to the equation modeling amino acid biosynthesis (**Equation 9**) is that it implies that both amino acids and total protein content scale with the number of ribosomes ultimately leading to a homeostatic dry mass density independent of time (see Appendix 1, section 4.1):

$$\rho^H = \frac{\mathcal{M}_a}{\frac{v_p}{l_p} + \frac{(z_{Af}+1)}{2n_0} \cdot \left( \frac{\phi_e}{l_p} \cdot \frac{k_{cat}}{k_r} - 1 \right)} \tag{10}$$

We emphasize that **Equation 10** only applies far from its singularity since it was obtained assuming that the volume of the cell is determined by free amino acids, that is, $\frac{\phi_e}{l_p} \cdot \frac{k_{cat}}{k_r} \gg 1$.

Not only does our model explain the homeostasis of the dry mass, but it also makes the salient prediction that this homeostasis naturally breaks down if the time spent in the G1 phase is too long. Indeed, after a time $t^{**} = \frac{1}{kr} \cdot ln\left( \frac{\mathcal{N}_r^{max} \cdot \mathcal{N}_p^{max} \cdot k_0 \cdot \tau_m \cdot \sum g_j}{P_{r,0}} \right)$ (see Appendix 1, section 4.1), mRNAs become saturated by ribosomes, drastically changing the growth of proteins from an exponential growth to a plateau regime where the number of proteins remains constant. After the time $t^{**} + \tau_p$, all protein numbers reach their stationary values $P_j^{stat} = k_t \cdot k_0 \cdot \tau_p \cdot \tau_m \cdot \mathcal{N}_R^{max} \cdot \mathcal{N}_p^{max} \cdot g_j$. In particular, the enzymes coding for amino acids also plateau implying the loss of the scaling between free amino acids and proteins as predicted by **Equation 9**. The number of amino acids then increases linearly with time, whereas the number of proteins saturates. In this regime, the volume thus grows linearly with time but the dry mass remains constant, leading to its dilution and the decrease in the dry mass density (see Appendix 1, section 4.1 and **Equation A.45**):

$$\rho^{lin}(t) = \frac{\mathcal{M}_a}{\frac{v_p}{l_p} + \frac{(z_{Af}+1)}{2n_0} \cdot \left( \frac{\phi_e \cdot k_{cat}}{\phi_r \cdot k_r \cdot k_t \cdot l_p \cdot \tau_p} - 1 + \frac{k_{cat} \cdot \phi_e}{l_p} \cdot t \right)} \tag{11}$$

Finally, our model makes other important predictions related to the cell ploidy that we briefly enumerate. First, the cut-off $P_r^*$ (**Equation A.41**) at which dilution is predicted to occur depends linearly on the genome copy number $\sum g_j$. Intuitively, mRNAs are saturated only if DNA has previously saturated. At saturation, the RNA number is proportional to the genome size. As a consequence, the volume $V^* \propto P_r^*$ at which dilution occurs scales with the ploidy of the cell, a tetraploid cell is predicted to be diluted at twice the volume of its haploid homolog. On the other hand, by virtue of the exponential growth, the time $t^{**}$ (**Equation A.42**) at which the saturation occurs only depends logarithmically on the number of gene copies making the ploidy dependence much less pronounced timewise. Second, the growth rate in the linear regime scales with the ploidy of the cell, as opposed to the growth rate in the exponential regime. Indeed, in the saturated regime, the growth rate scales as $k_{cat} \cdot P_e^{stat}$ (see Appendix 1, section 4.1 and **Equation A.43**), where $P_e^{stat}$ is the number of enzymes catalyzing the reaction of amino acids biosynthesis after their numbers have reached their stationary values, while in the exponential regime, the growth rate $k_r = \phi_r \cdot k_t - \frac{1}{\tau_p}$ scales with the fraction of genes coding for ribosomes $\phi_r$, which is independent of the ploidy.

## Comparison to existing data

Our main prediction, namely that the cell is diluted after the end of the exponential growth, is reminiscent of the intracellular dilution at senescence recently reported in fibroblasts, yeast cells, and more recently suspected in aged hematopoietic stem cells (*Neurohr et al., 2019*; *Lengefeld et al., 2021*). Here we quantitatively confront our theory to the data of *Neurohr et al., 2019*, where the volume, the dry mass, and the protein number were recorded during the growth of yeast cells that were prevented from both dividing and replicating their genome. Though our theory was originally designed for mammalian cells, it can easily be translated to cells with a cell wall provided that the hydrostatic pressure difference across the wall $\Delta P$ is maintained during growth by progressive incorporation of cell wall components (see Appendix 1, section 2.1.2). Indeed, our conclusions rely on the fact that the cell volume is primarily controlled by small osmolytes whose concentration in the cell dominates the osmotic pressure, a feature observed to be valid across cell types (*Park et al., 2016*).

We first check the qualitative agreement between our predictions and the experiments. Two distinct growth regimes are observed, at least on the population level, in the volume data of nondividing yeast cells (*Neurohr et al., 2019*) an initial exponential growth followed by a linear growth (*Figure 2C*). The occurrence of linear or exponential growth has been the object of intense debate. One of the ambiguity may come from the fact that cells often divide before there is a clear distinction between the linear and the exponential regimes ($e^t \approx 1 + t + \mathcal{O}(t^2)$, for $t << 1$). Note that our model only requires the exit of the exponential growth regime to observe the dilution. Moreover, our results suggest that both regimes of growth are not equivalent for proper cell function. Cells that would remain in the 'nonexponential' growth regime for too long will be diluted. This may be at the root of the observed cell cycle defects (*Neurohr et al., 2019*) and be the cause of functional decline towards senescence in particular for fibroblasts (*Neurohr et al., 2019*) and hematopoietic stem cells (*Lengefeld et al., 2021*). We think that a more precise understanding of the relationship between dilution and cell cycle defects remains an exciting avenue for future research. Our theory also predicts that as long as protein number is constant the volume must grow linearly (*Equations 9 and A.45*). This is precisely what is observed in the experiments: cells treated with rapamycin exhibit both a constant protein content and a linear volume increase during the whole growth (see Figure S6.F in *Neurohr et al., 2019*). Finally, our predictions on the relationship between ploidy and dilution are in very good agreement with experiments as well. Indeed, while the typical time to reach the linear growth regime – of the order of 3 hr – seems independent of the ploidy of the cell, the volume at which dilution occurs is doubled (see Figure S7.A in *Neurohr et al., 2019*). Moreover, the growth rate during the linear regime scales with ploidy as the haploid cells growth rate is of order 129 fL/h against 270 fL/h for their diploid counterparts (*Neurohr et al., 2019*).

Encouraged by these qualitative correlations, we further designed a scheme to test our theory more quantitatively. Although our theory has a number of adjustable parameters, many of them can be combined or determined self-consistently as shown in Appendix 1, section 4.1.4. We end up fitting four parameters, namely $\tau_p$, $t^{**}$, $k_r$ and the initial cell volume $v_1$, using the cell volume data (*Figure 2B*). We detail in Appendix 1, section 4.1.5 the fitting procedure and the values of the optimal parameters. Interestingly, we find a protein degradation time $\tau_p = 1\,\text{hr}\,9\,\text{min}$, corresponding to an average protein half-life time: $\tau_{1/2} \sim 48\,\text{min}$, which is very close to the value 43 min, measured in *Belle et al., 2006*. Moreover, we obtain a saturation time $t^{**} = 2\,\text{hr}\,44\,\text{min}$, which remarkably corresponds to the time at which the dry mass density starts to be diluted (*Figure 2E*), thus confirming the most critical prediction of our model. We can then test our predictions on the two other independent datasets at our disposal, that is, the dry mass density, obtained from suspended microchannel resonator (SMR) experiments, and the normalized protein number, from fluorescent intensity measurements. We emphasize that the subsequent comparisons with experiments are done without any adjustable parameters. The agreement between theory and experiment is satisfactory (*Figure 2D and E*) and gives credit to our model. We underline that the value of the density of water that we used is 4% higher than the expected value, $\rho_w = 1.04\,\text{kg/L}$ to plot *Figure 2E*. This slight difference originates from the fact that our simplified theory assumes that the dry mass is entirely due to proteins whereas proteins represent only 60% of the dry mass. This hypothesis is equivalent to renormalizing the density of water as shown in Appendix 1, section 4.1.4.

In summary, our theoretical framework combining the Pump-Leak model with a growth model and a model of amino acid biosynthesis provides a consistent quantitative description of the dry mass

density homeostasis and its subsequent dilution at senescence without invoking any genetic response of the cell; the dilution is due to the physical crowding of mRNAs by ribosomes. It also solves a seemingly apparent paradox stating that the volume is proportional to the number of proteins although their concentrations are low in the cell without invoking any nonlinear term in the osmotic pressure (see 'Discussion' and Appendix 1, section 3.1.12).

## Mitotic swelling

Our previous results explain well the origin of the dilution of the cellular dry mass at senescence. But can the same framework be used to understand the systematic dry mass dilution experienced by mammalian cells at mitotic entry? Although this so-called mitotic swelling or mitotic overshoot is believed to play a key role in the change of the physicochemical properties of mitotic cells, its origin remains unclear (*Son et al., 2015*; *Zlotek-Zlotkiewicz et al., 2015*).

We first highlight five defining features of the mitotic overshoot. (1) It originates from an influx of water happening between prophase and metaphase, resulting in a typical 10% volume increase in the cells. (2) The swelling is reversible and cells shrink back to their initial volume between anaphase and telophase. (3) This phenomenon appears to be universal to mammalian cells, larger cells displaying larger swellings. (4) Cortical actin was shown not to be involved in the process, discarding a possible involvement of the mechanosensitivity of ion channels, contrary to the density increase observed during cell spreading (*Venkova et al., 2022*) (5) Nuclear envelope breakdown (NEB) alone cannot explain the mitotic overshoot since most of the swelling is observed before the prometaphase where NEB occurs (*Son et al., 2015*; *Zlotek-Zlotkiewicz et al., 2015*).

The dry mass dilution at mitotic overshoot is thus different from the cases studied in the previous section. First, it happens during mitosis when the dry mass is constant (*Zlotek-Zlotkiewicz et al., 2015*). Second, the 10% volume increase implies that we need to improve the simplified model used above, which considers only metabolites and proteins (and their counterions). Having in mind that ions play a key role in the determination of the cell volume (*Figure 1B*), we show how every feature of the mitotic overshoot can be qualitatively explained by our theory, based on a well-known electrostatic property of charged polymer called counterion condensation first studied by *Manning, 1969*. Many counterions are strongly bound to charged polymers (such as chromatin) because the electrostatic potential at their surface creates an attractive energy for the counterions much larger than the thermal energy $k_B T$. The condensed counterions partially neutralize the charge of the object and reduce the electrostatic potential. Condensation occurs up to the point where the attractive energy for the free counterions is of the order $k_B T$. The condensed counterions then do not contribute to the osmotic pressure given by *Equation 2*, which determines the cell volume. These condensed counterions act as an effective 'internal' reservoir of osmolytes. A release of condensed counterions increases the number of free cellular osmolytes and thus the osmotic pressure inside the cell. Therefore, it would lead to an influx of water in order to restore osmotic balance at the plasma membrane (*Figure 3*).

But how to explain such a counterion release at mitotic overshoot? For linear polymers such as DNA, the condensation only depends on a single Manning parameter $u = \frac{l_b}{A}$; where $l_b$ is the Bjerrum length (*Appendix 1—table 1*) that measures the strength of the coulombic interaction and $A$ the average distance between two charges along the polymer. The crucial feature of Manning condensation is the increase in the distance between charges $A$ by condensing counterions and thus effectively decreasing $u$ down to its critical value equal to $u^{eff} = 1$ (see Appendix 1, section 5.1 for a more precise derivation). Hence, the number of nominal elementary charges carried by a polymer of length $L_{tot}$ is $Q_{tot} = \frac{L_{tot}}{A}$. Due to condensation, the effective distance between charges increases to $A^{eff} = l_b$ such that the effective number of charges on the polymer is reduced to $Q^{eff} = \frac{Q_{tot}}{u}$. The number of counterions condensed on the polymer is $Q^{cond} = Q_{tot} \cdot (1 - \frac{1}{u})$. The most important consequence of these equations is that they suggest that a structural modification of the chromatin could lead to a counterion release. Indeed, making the chromatin less negatively charged, that is, increasing $A$, is predicted to decrease $u$ and thus to lead to the decrease in $Q^{cond}$. Detailed numerical simulations of chromatin electrostatics show that this description is qualitatively correct (*Materese et al., 2009*).

Biologists have shown that chromatin undergoes large conformational changes at mitotic entry. One of them attracted our attention in light of the mechanism that we propose. It is widely accepted that the affinity between DNA and histones is enhanced during chromatin compaction by stronger electrostatic interactions thanks to specific covalent modifications of histone tails by enzymes. Some

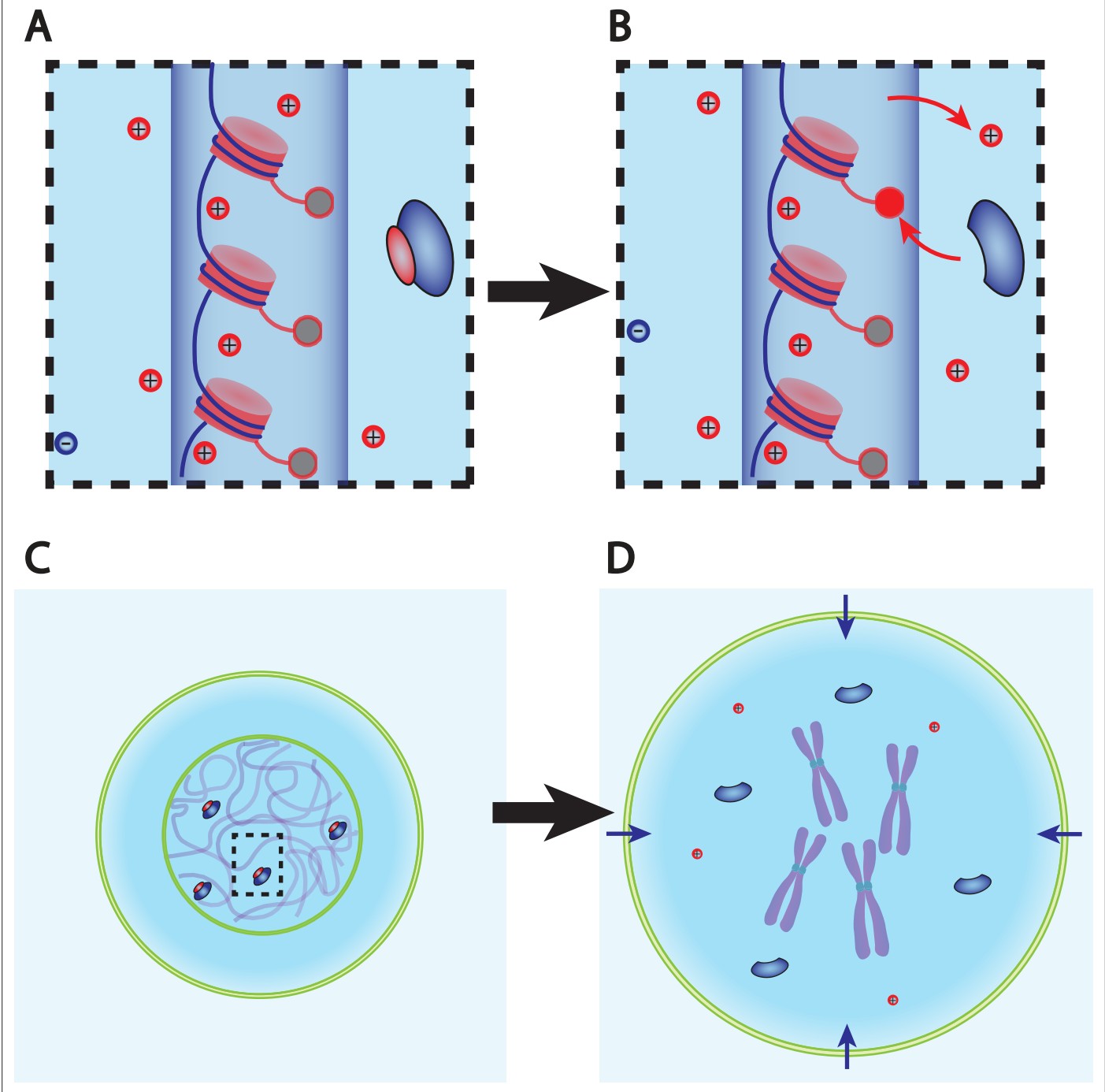

**Figure 3.** Dry mass dilution at mitosis is explained with the Pump-Leak model by the decondensation of chromatin counterions following histone tail modifications. (**A, B**) Microscopic working model. An enzyme gives its positive charge to a histone, resulting in the release of a condensed counterions. Ions depicted within the chromatin (dark blue cylinder) are condensed and those outside are freely diffusing and participate in the nuclear osmotic pressure. (**C, D**) The subsequent increase in the number of osmolytes lead to a water influx in order to sustain osmotic balance at the plasma membrane of mammalian cells. For readability, other osmolytes are not displayed.

of these modifications such as the deacetylation of lysines add a positive charge to the histone tails, hence making the chromatin less negatively charged (*Alberts et al., 2002*). Moreover, histone tails are massively deacetylated during chromatin compaction (*Zhiteneva et al., 2017*), potentially meaning that this specific reaction plays an important role in counterion release and thus on the observed mitotic swelling. However, we underline that the idea that we propose is much more general and

that any reaction modifying chromatin electrostatics is expected to impact the swelling. The question whether deacetylation of lysines is the dominant effect is left open here.

Is the proposed mechanism sufficient to explain the observed 10% volume increase? We estimate the effective charge of chromatin for a diploid mammalian cell to be $Q^{eff} = 2 \cdot 10^9 \, e^-$ and the number of condensed monovalent counterions to be $Q^{cond} = 8 \cdot 10^9$ (see Appendix 1, section 3.1.6 and 3.1.7). The Pump-Leak model framework predicts the subsequent volume increase induced by the hypothetical release of all the condensed counterions of the chromatin. We find an increase of order $\Delta V \sim 100 - 150 \mu m^3$, which typically represents 10% of a mammalian cell size (see Appendix 1, section 3.1.8 and *Equation A.29*). Admittedly crude, this estimate suggests that chromatin counterion release can indeed explain the amplitude of mitotic swelling.

In summary, the combination of the Pump-Leak model framework with a well-known polymer physics phenomenon allows us to closely recapitulate the features displayed during mitotic swelling. In brief, the decondensation of the chromatin condensed counterions, hypothetically due to histone tail modifications, is sufficient to induce a 10% swelling. This implies that all mammalian cells swell during prophase and shrink during chromatin decondensation after anaphase; again, consistent with the dynamics of the mitotic overshoot observed on many cell types. Another salient implication is that the amplitude of the swelling is positively correlated with the genome content of the cells: cells having more chromatin are also expected to possess a larger 'internal reservoir' of osmolytes, which can participate in decondensation. This provides a natural explanation for the observed larger swelling of larger cells. For instance, Hela cells were shown to swell on average by 20%, in agreement with the fact that many of them are tetraploid. Admittedly, many other parameters enter into account and may disrupt this correlation such as the degree of histone tail modifications or the initial state of chromatin; The existence of a larger osmolyte reservoir does not necessarily mean that more ions are released.

Finally, we point out that the ideas detailed in this section can be tested experimentally using existing in vivo or in vitro methods. For example, we propose to massively deacetylate lysines during interphase by either inhibiting lysine acetyltransferases (KATs) or overexpressing lysine deacetylases (HDACS) in order to simulate the mitotic swelling outside mitosis. We also suggest to induce mitotic slippage or cytokinesis failure for several cell cycles, to increase the genome content, while recording the amplitude of swelling at each entry in mitosis (*Gemble et al., 2022*).

## Nuclear scaling

Another widely documented scaling law related to cell volume states that the volume of cell organelles is proportional to cell volume (*Chan and Marshall, 2010*; *Cantwell and Nurse, 2019b*). As an example, we discuss here the nuclear volume. We develop a generalized 'nested' Pump-Leak model that explicitly accounts for the nuclear and plasma membranes (see *Figure 1A*). Instead of writing one set of equation (*Equations 1–3*) between the interior and the exterior of the cell, we write the same equations both inside the cytoplasm and inside the nucleus (see *Equation A.55*). Before solving this nonlinear system of equations using combined numerical and analytical approaches, we draw general conclusions imposed by their structure. As a thought experiment, we first discuss the regime where the nuclear envelope is not under tension so that the pressure jump at the nuclear envelope $\Delta P_n$ is much smaller than the osmotic pressure inside the cell $\Delta P_n << \Pi_0$. The osmotic balance in each compartment implies that the two volumes have the same functional form as in the Pump-Leak model, with two contributions: an excluded volume due to dry mass and a wet volume equal to the total number of particles inside the compartment divided by the external ion concentration (see *Equation A.56*). It is noteworthy that the total cell volume, the sum of the nuclear and cytoplasmic volumes, is still given by *Equation 4* as derived in the simple Pump-Leak model. This result highlights the fact that the Pump-Leak model strictly applies in the specific condition where the nuclear envelope is under weak tension. In addition, a crucial consequence of the osmotic balance condition at the nuclear envelope is that it leads to a linear scaling relation between the volumes of the two compartments:

$$V_n = \frac{N_n^{tot}}{N_c^{tot}} \cdot V_c + \left( R_n - \frac{N_n^{tot}}{N_c^{tot}} \cdot R_c \right) \tag{12}$$

where $V_i$, $R_i$, and $N_i^{tot}$ denote, respectively, the total volume, dry volume, and total number of osmolytes of compartment $i$, the index $i = n, c$ denoting either the nucleus, $n$, or the cytoplasm, $c$. Importantly, this linear scaling between the nucleus and the cytoplasm was reported repeatedly over the last century and is known as nuclear scaling (*Webster et al., 2009*; *Cantwell and Nurse, 2019b*).

While this conclusion is emphasized in some recent papers (*Lemière et al., 2022*; *Deviri and Safran, 2022*), we point out that *Equation 12* is only a partial explanation of the robustness of the nuclear-scaling law. To further understand this affirmation and also to motivate our work, we first consider the simpler case of a cell containing proteins, chromatin, and their counterions (no metabolites). For the sake of readability, we assume that the volume fraction occupied by the dry mass is the same in the nucleus and in the cytoplasm (see Appendix 1, section 6.1.1). The NC ratio is then given by the ratio of the wet volumes. The osmotic balance at the nuclear envelope reads

$$(z_p + 1) \cdot p_n + q^{eff} = (z_p + 1) \cdot p_c \quad (13)$$

where $z_p$ is the average charge of proteins, $q^{eff}$ is the concentration of the counterions of chromatin, and $p$ is the protein concentration either in the nucleus, subscript $n$, or in the cytoplasm, subscript $c$. The term $z_p \cdot p_n$ hence accounts for the concentration of counterions associated to the proteins trapped in the nucleus. The NC ratio can thus be expressed as

$$\frac{V_n}{V_c} = \frac{(z_p+1) \cdot P_n + Q^{eff}}{(z_p+1) \cdot P_c} \quad (14)$$

where the capital letters $P_i$ and $Q^{eff}$ now account for the number of proteins and chromatin counterions. It is noteworthy that although being permeable to the nuclear envelope, ions can still play a role in the NC ratio. A flawed reasoning would state that their permeability implies that their concentration is balanced at the nuclear envelope, and hence that their contribution to the NC ratio can be discarded. Here, the condition of electroneutrality inside the nucleus leads to the appearance of a nuclear difference of potential that effectively traps the ions inside the nucleus and thus creates an imbalance of ion concentration. Of course, this effect would be negligible if $Q^{eff} << (z_p + 1) \cdot P_n$. However, our estimate goes against this hypothesis as $Q^{eff} \sim (z_p + 1) \cdot P_n$ for both mammalian and yeast cells (see Appendix 1, sections 3.1.6 and 3.1.10). The problem that arises is that $Q^{eff}$, which we remind is not negligible here, does not scale during growth. This would imply that the NC ratio decreases during growth (see *Figure 4*). We solve this apparent paradox in the next section by considering metabolites; a consideration that has largely been overlooked in the recent literature (*Wu et al., 2022*; *Lemière et al., 2022*; *Deviri and Safran, 2022*).

## Role of metabolites on the NC ratio in the low tension regime

We now examine the influence of the metabolites on the NC ratio. Following the lines of our previous discussion, four different components play a role in volume regulation: chromatin (indirectly through its noncondensed counterions), proteins (mainly contributing to the dry volume), and metabolites and ions (mainly contributing to the wet volume). It is noteworthy that these components do not play symmetric roles in the determination of the NC ratio. This originates from the fact that metabolites are permeable to the nuclear membrane and that chromatin, considered here as a polyelectrolyte gel, does not contribute directly to the ideal gas osmotic pressure because its translational entropy is vanishingly small (*de Gennes, 1979*). The nested Pump-Leak model leads to highly nonlinear equations that cannot be solved analytically in the general case (see *Equation A.55*). Nevertheless, in the particular regime of monovalent osmolytes and high pumping $z_a = 1$, $z_p = 1$ and $\alpha_0 = 0$ corresponding to the case where there is no free anions in the cell, the equations simplify and are amenable to analytical results. This regime is physically relevant since it corresponds to values of the parameters close to the ones that we estimated (*Table 1*). For clarity, we first restrict our discussion to this particular limit. We will also discuss both qualitatively and numerically the influence of a change of the parameters later. In this scenario, the nested Pump-Leak model equations reduce to

$$\begin{cases} p_c + a_c^f + n_c = 2n_0 \\ p_n + a_n^f + n_n = 2n_0 \\ n_c - a_c^f - p_c = 0 \\ n_n - a_n^f - p_n - q^{eff} = 0 \\ n_c \cdot a_c^f = n_n \cdot a_n^f \end{cases} \quad (15)$$

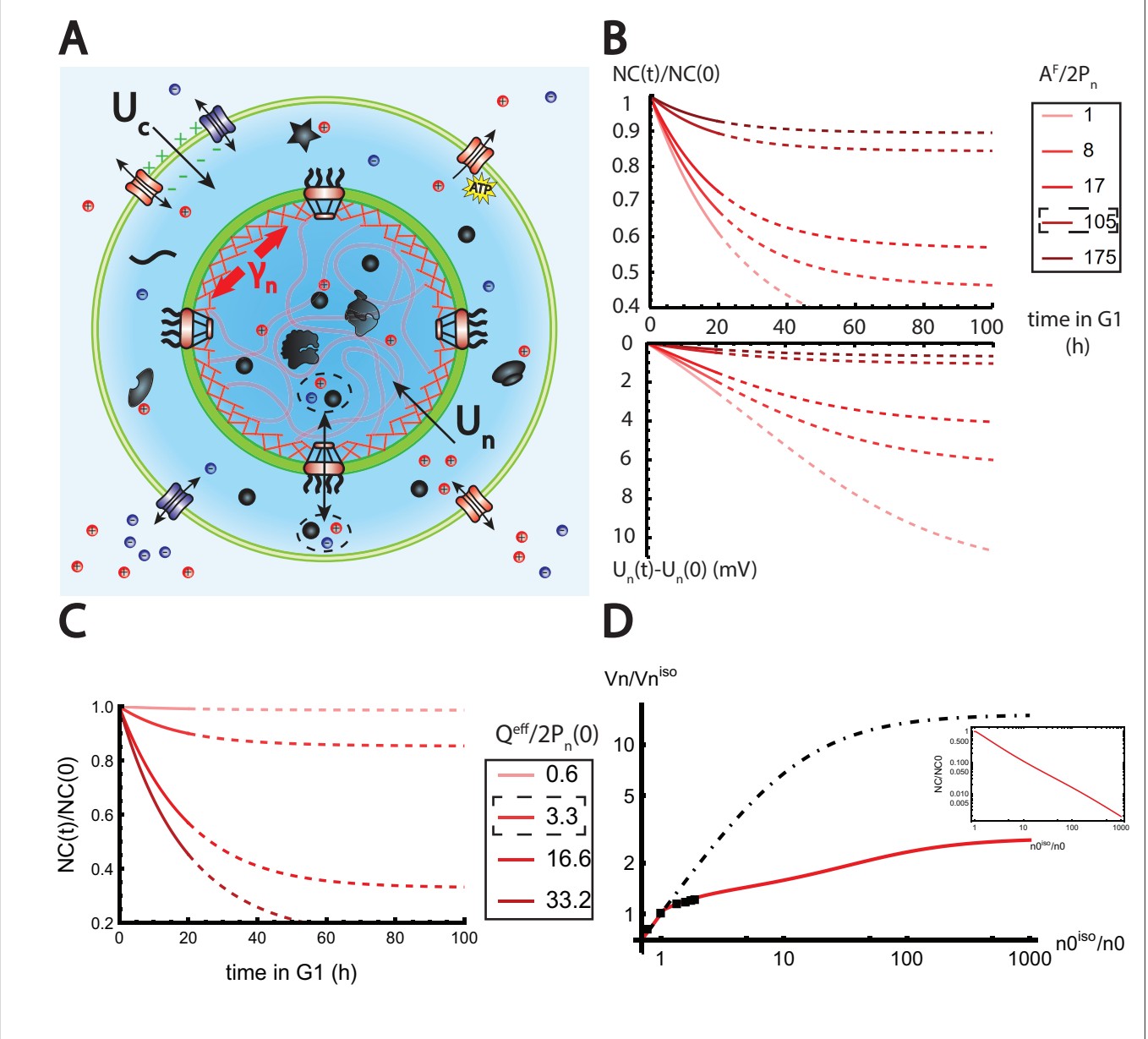

**Figure 4.** The nested Pump-Leak model explains nuclear scaling. (**A**) Schematic of the nested Pump-Leak model. Species in black are impermeant molecules (X) and are now partitioned between the cytoplasm and the nucleus. Among those, only metabolites (black circles) can cross the nuclear envelope. The nuclear envelope is composed of the membrane (green) and the lamina (red) can be stretched when the nuclear folds are flattened. (**B, C**) Simulations of the nested Pump-Leak model (*Equation A.55*) during growth when the osmotic pressure is balanced at the nuclear envelope. The growth rate was adjusted to data in *Zlotek-Zlotkiewicz et al., 2015* (**B**) Though permeable to the nuclear envelope, metabolites play a role in the homeostasis of the nuclear-to-cytoplasmic (NC) ratio by diluting chromatin (free) counterions which do not scale during growth (top plot). Higher variations of the NC ratio correlate with higher variations of the nuclear envelope potential (bottom plot). (**C**) Variations of the NC ratio during growth for different chromatin charges. (**D**) Normalized nuclear volume after a hypo-osmotic shock. Nuclear volume saturates because of the tension at the nuclear envelope, leading to the decrease in the NC ratio (inset: log–log plot). The dash-dotted line represents the nuclear volume if the number of osmolytes in the nucleus were assumed constant throughout the shock. Thus, showing that metabolites leave the nucleus during the shock which strongly decreases nucleus swelling. The value at the saturations are given by *Equation 19*. The square black dots are data extracted from Figure 3 from *Finan et al., 2009*. We used $K = 50mN/m$ and $s = 4\%$ folds to fit the data. The envelope stretching modulus $K$ used is twice the measured value in *Dahl et al., 2004*. The rationale is threefold. (1) Nuclei used in *Finan et al., 2009* are chondrocyte nuclei originating from articular cartilage. They possess a high lamina A to lamina B ratio and are thus likely to be stiffer (*Swift et al., 2013*). (2) We could lower the value of the fitted $K$ by increasing the pumping efficiency $\alpha_0$. A more detailed characterization of the Pump-Leak model parameters for chondrocytes would be required to precisely infer the elastic properties of the nuclear envelope. (3) Considering the chromatin mechanical contribution would increase $K$ by a factor $E_{DNA} \cdot R_{nucleus}$; with $E_{DNA}$ the elastic modulus of the chromatin and $R_{nucleus}$ the radius of the nucleus. Additional results of the nested Pump-Leak model are plotted in

*Figure 4 continued*

**Appendix 1—figure 1**. The online version of this article includes the following source data for *Figure 1*: *Figure 4—source data 1*. Data table extracted from *Finan et al., 2009*.

The online version of this article includes the following source data for figure 4:

**Source data 1.** Data table exctracted from Figure 3 of *Finan et al., 2009*.

---

where the first and second equations correspond to osmotic pressure balance in the two compartments; the third and fourth equations correspond to macroscopic electroneutrality in each compartment; and the fifth equation is the balance of the chemical potential of the cations and metabolites on each side of the nuclear envelope. $p_i, n_i, d_i^f$ respectively account for the concentrations of proteins, cations, and metabolites either in the cytoplasm – subscript $c$ – or in the nucleus, subscript $n$. $q^{eff}$ accounts for the effective chromatin charge density. From these equations, we express the concentrations of cations in each compartment as functions of the extracellular concentration $n_0$ and the chromatin charge density $q^{eff}$ (*Equation A.60*), leading to the following expression of the nuclear envelope potential:

$$U_n = -ln\left(1 + \frac{q^{eff}}{2n_0}\right) = -ln\left(1 + \frac{Q^{eff}}{Q^{eff} + 2A_n^f + 2P_n}\right) \tag{16}$$

A salient observation from *Equation 16* is that the nuclear envelope potential difference $U_n$ is a proxy of the chromatin charge density. At low $q^{eff}$, $U_n = 0$, that is, the respective concentrations of metabolites and cations are equal on each side of the membrane. *Equation 15* also shows that the protein concentrations are equal in the two compartments. This implies that when the charge of chromatin is diluted, the volumes of the nucleus and of the cytoplasm adjust such that the NC ratio equals the ratio of protein numbers in the two compartments $NC_1 = \frac{P_n}{P_c}$. In the Pump-Leak model, which considers a single compartment, a membrane potential appears as soon as there exist trapped particles in the compartment (see Appendix 1, section 6.1.2 and *Equation A.58*). In contrast, our extended nested Pump-Leak model predicts that in the case of two compartments, the system has enough degrees of freedom to adjust the volumes as long as $q^{eff}$ is small, thereby allowing the potential to be insensitive to the trapped charged proteins. At high values of the chromatin charge $Q^{eff}$, $U_n$ saturates to the value $-ln(2)$, which in physical units is equivalent to $-17\,\mathrm{mV}$ at $300\,\mathrm{K}$. Note that this lower bound for the potential is sensitive to the average charge of the proteins $z_p$ and can be lowered by decreasing this parameter. We also highlight that *Equation 16* makes another testable prediction, namely, that the nuclear envelope potential is independent of the external ion concentration. In the literature, nuclear envelope potentials were recorded for several cell types (*Mazzanti et al., 2001*). They can vary substantially between cell types ranging from $\sim 0\,\mathrm{mV}$ for *Xenopus* oocytes to $-33\,\mathrm{mV}$ for Hela cells. This result is in line with our predictions. The *Xenopus* oocyte nucleus has a diameter roughly 20 times larger than typical somatic nuclei, but its chromatin content is similar (*Dahl et al., 2004*), resulting in a very diluted chromatin and a vanishing nuclear envelope potential. On the other hand, Hela cells are known to exhibit an abnormal polyploidy that may lead to a large chromatin charge density and a large nuclear membrane potential.

This last observation allows to understand the influence of the metabolites on the NC ratio. An increase in the number of metabolites in the cell $A_{tot}^f$ induces growth of the total volume (*Equation A.56*), leading to the dilution of the chromatin charge and a strong decrease in the nuclear membrane potential (*Equation 16*). In the limit where $A_{tot}^f$ is dominant, we thus expect the NC ratio to be set to the value $NC_1$. On the other hand, at low $A_{tot}^f$, metabolites do not play any role on the NC ratio, which is then given by $NC_2 > NC_1$ (see *Equation A.59* for the general formula), with

$$NC_1 = \frac{P_n}{P_c} \quad , \qquad NC_2 = \frac{P_n + Q^{eff}/2}{P_c} \tag{17}$$

The actual NC ratio is intermediate between the two limiting behaviors (see *Appendix 1—figure 1B* and *Equation A.65*). Note that the regime $NC_2$ is equivalent to the regime given by *Equation 14* found in our preliminary discussion.

During cell growth, the ratio $NC_1$ is constant, while the ratio $NC_2$ varies with time. Indeed, if nucleo-cytoplasmic transport is faster than growth, the protein numbers $P_n$ and $P_c$ are both proportional to the number of ribosomes in the exponential growth regime and the ratio $NC_1$ does not vary

with time (see Appendix 1, section 6.1.5). On the other hand, the DNA charge $Q^{eff}$ is constant during G1 phase while $P_n$ grows with time, so $NC_2$ decreases with time. The fact that the NC ratio remains almost constant during growth (*Neumann and Nurse, 2007*; *Pennacchio et al., 2022*) suggests that cells are closer to the $NC_1$ regime and point at the crucial role of metabolites in setting the NC ratio (*Figure 4* and *Appendix 1—figure 1B*). Importantly, these conclusions are overlooked in a large part of the existing literature that often assumes that metabolites do not play any role on the NC ratio due to their permeability at the nuclear envelope. We end this qualitative discussion by predicting the effect of a variation of the parameters $z_p, z_a$ and $\alpha_0$ that were so far assumed to be fixed. Our main point is that any parameter change that tends to dilute the chromatin charge also tends to increase the (negative) nuclear envelope potential and make the NC ratio closer to the regime $NC_1$ and farther from the regime $NC_2$. Consequently, when $z_p$ or $z_a$ are increased, the number of counterions carried by each protein or metabolite increases. This in turn results in a global growth of the volume and hence leads to the dilution of the chromatin charge and to the increase in the nuclear envelope potential difference. Any increase in the pumping parameter $\alpha_0$ (decrease in pumping efficiency) has a similar effect. It increases the number of ions in the cell, resulting again in the dilution of the chromatin charge. Note that in the absence of pumping ($\alpha_0 = 1$), the Pump-Leak model predicts a diverging volume because this is the only way to enforce the balance of osmotic pressures at the plasma membrane (*Equation A.11*) (if there is no pressure difference at the membrane due to a cell wall).

Five crucial parameters have emerged from our analytical study: (1) $\frac{P_n}{P_c}$ ; (2) $\frac{A^f}{2P_n}$; (3) $\frac{Q^{eff}}{2P_n}$; (4) $\alpha_0$ ;and (5) $z_p$ and $z_a$. But what are the biological values of these parameters? We summarize our estimates in Appendix 1, section 3.1. Importantly, the ratio between chromatin (free) counterions and the number of nuclear trapped proteins (and their counterions) is estimated to be of order one (see Appendix 1, section 3.1 and *Figure 4C*). As a key consequence, we find that the NC ratio would be four times larger in the absence of metabolites (see *Appendix 1—figure 1B*). This nonintuitive conclusion sheds light on the indirect, yet fundamental, role of metabolites on the NC ratio, which have been overlooked in the literature.

We now turn to a numerical solution to obtain the normalized variations of the NC ratio during growth in the G1 phase for different parameters (*Figure 4*). Interestingly, variations of the NC ratio and variations of the nuclear envelope potential are strongly correlated, a feature that can be tested experimentally (*Figure 4B*). Moreover, we deduce from our numerical results that, in order to maintain a constant NC ratio during the cell cycle, cells must contain a large pool of metabolites (see *Figure 4C*). Our estimates point out that this regime is genuinely the biological regime throughout biology, thus providing a natural explanation on the origin of the nuclear scaling.

In summary, many of the predictions of our analysis can be tested experimentally. Experiments tailored to specifically modify the highlighted parameters are expected to change the NC ratio. For example, we predict that depleting the pool of metabolites by modifying amino acid biosynthesis pathways, that is, lowering $\frac{A^f}{2P_n}$, would lead to an increase in the NC ratio. Importantly, good metabolic targets in these experiments could be glutamate or glutamine because they account for a large proportion of the metabolites in the cell (*Park et al., 2016*). We also point out that cells with a smaller metabolic pool are expected to experience higher variations of the NC ratio during growth and thus larger fluctuations of this ratio at the population level (*Figure 4B*). These predictions could shed light on understanding the wide range of abnormal karyoplasmic ratio among cancer cells. Indeed, metabolic reprogramming is being recognized as a hallmark of cancer (*Fujita et al., 2020*) some cancer cells increase their consumption of the pool of glutamate and glutamine to fuel the TCA cycle and enhance their proliferation and invasiveness (*Altman et al., 2016*).

Moreover, disruption of either nuclear export or import is expected to change $\frac{P_n}{P_c}$ and thus the NC ratio. Numerical solutions of the equations displayed in *Appendix 1—figure 1* show a natural decrease of the NC ratio due to the disruption of nuclear import. On the other hand, if nuclear export is disrupted, we expect an increase in the NC ratio. This is in agreement with experiments done recently in yeast cells (*Lemière et al., 2022*). The authors reported a transient decrease followed by an increase in the diffusivities in the nucleus. This is in line with what our theory would predict. The initial decay is due to the accumulation of proteins in the nucleus, resulting in an associated crowding. On the other hand, the following increase is due to the impingement of ribosome synthesis as this

step requires nuclear export. Our model would then predict the loss of the exponential growth and a decoupling between protein and amino acid numbers that would drive the dilution of the nuclear content.

Finally, our framework also predicts that experiments that would maintain the five essential parameters unchanged would preserve the nuclear scaling. We thus expect that, as long as the nuclear envelope is not under strong tension, changing the external ion concentration does not influence the scaling directly. Experiments already published in the literature (*Guo et al., 2017*) show precisely this feature.

## Mechanical role of the lamina on the NC ratio

So far we have assumed that the osmotic pressure is balanced at the nuclear envelope, which is a key condition for the linear relationship between nuclear and cytoplasmic volume. But why should this regime be so overly observed in biology? We first address this question qualitatively. For simplicity in the present discussion, we assume that DNA is diluted so that the nuclear envelope potential is negligible. This implies that metabolites and ions are partitioned so that their concentrations are equal in the nucleoplasm and the cytoplasm, hence canceling their contribution to the osmotic pressure difference at the nuclear envelope. This allows to express the volume of the nucleus as

$$V_n = \frac{P_n}{\frac{\Delta P_n}{k_B T} + p_c}$$

(18)

While the previous expression is not the exact solution of the equations, it qualitatively allows us to realize that the nuclear envelope hydrostatic pressure difference plays a role in the volume of the nucleus if it is comparable to the osmotic pressure exerted by proteins. This pressure is in the 1000 Pa range since protein concentration are estimated to be in the millimolar range (Appendix 1, section 3.1). We further estimated an upper bound for the nuclear pressure difference to be in the $10^4$ Pa range (*Equation A.33*). Admittedly crude, these estimates allow us to draw a threefold conclusion. (1) The nuclear pressure difference $\Delta P_n$ can be higher than the cytoplasmic pressure difference $\Delta P_c$, in part due to the fact that lamina has very different properties compared to cortical actin: it is much stiffer and its turnover rate is lower. This points out the possible role of nuclear mechanics in the determination of the nuclear volume contrary to the cortical actin of mammalian cells that does not play any direct role for the cell volume. (2) The typical hydrostatic pressure difference at which mechanical effects become relevant is at least two orders of magnitude lower for the nucleus than for the cytoplasm, for which it is of order $\Pi_0$, (3) Assuming linear elasticity, small nuclear envelope extensions of 10% would be sufficient to impact nuclear volume. These conclusions stand in stark contrast to the observed robustness of the nuclear scaling, thus pointing out that the constitutive equation for the tension in the lamina is nonlinear. Biologically, we postulate that this nonlinearity originates from the folds and wrinkles that many nuclei exhibit (*Lomakin et al., 2020*). These folds could indeed play the effective role of membrane reservoirs, preventing the nuclear envelope tension to grow with the nuclear volume, hence setting the nuclear pressure difference to a small constant value, and maintaining cells in the scaling regime discussed in the previous sections. This conclusion is consistent with the results of *Finan et al., 2009*, which observed that the nucleus exhibits nonlinear osmotic properties.

To further confirm our conclusions quantitatively, we consider the thought experiment of nonadhered cells experiencing hypo-osmotic shock. This experiment is well adapted to study the mechanical role of nuclear components on nuclear volume because it tends to dilute the protein content while increasing the hydrostatic pressure by putting the nuclear envelope under tension. For simplicity, we ignore the mechanical contribution of chromatin that was shown to play a negligible role on nuclear mechanics for moderate extensions (*Stephens et al., 2017*). To gain insight into the nonlinear set of equations, we split the problem into two parts. First, we identify analytically the different limiting regimes of nuclear volume upon variation of the number of impermeant molecules $X_n$ present in the nucleus and the nuclear envelope tension $\gamma_n$. We summarize our results in a phase portrait (see Appendix 1, section 6.1.6 and *Appendix 1—figure 1*). Two sets of regimes emerge: those, studied above, where nuclear and cytoplasmic osmotic pressures are balanced, and those where the nuclear hydrostatic pressure matters. In the latter situations, the nuclear volume does not depend on the external concentration and saturates to the value (see Appendix 1, section 6.1.7):

$$\frac{V_n^{max}}{V_n^{iso}} = \frac{(1+s)^{3/2}}{2\sqrt{2}} \cdot \left(1 + \sqrt{1 + \frac{1}{(1+s)\cdot K^{eff}}}\right)^{3/2} \quad \text{with,} \quad K^{eff} = \frac{K}{k_B T \cdot \frac{N_n^{tot}}{V_n^{iso}} \cdot \left(\frac{6}{\pi} \cdot V_n^{iso}\right)^{1/3}} \tag{19}$$

where $s$ and $V_n^{iso}$ are respectively the fraction of membrane stored in the folds and the volume of the nucleus at the isotonic external osmolarity $2 \cdot n_0^{iso}$ is an effective adimensional modulus comparing the stretching modulus of the nuclear envelope $K$ with an osmotic tension that depends on the total number of free osmolytes contained by the nucleus $N_n^{tot}$. The saturation of the nuclear volume under strong hypo-osmotic shock originating from the pressure build up in the nucleus after the unfolding of the folds implies a significant decrease in the NC ratio and a loss of nuclear scaling (**Figure 4D**).

As a second step, we investigate the variations of $X_n = A_n^f + P_n$ after the shock. Our numerical solution again highlights the primary importance of considering the metabolites $A_n^f$ for the modeling of nuclear volume. Indeed, disregarding their contribution would lead to an overestimation of the number of trapped proteins. Additionally, $X_n$ would remain constant during the osmotic shock, resulting in the reduction of the effective modulus of the envelope (**Equation 19**). We would thereby overestimate the nuclear volume (**Figure 4D**, dashed line). In reality, since free osmolytes are mainly accounted for by metabolites that are permeable to the nuclear envelope, the number of free osmolytes in the nucleus decreases strongly during the shock. This decrease can easily be captured in the limit where metabolites are uncharged $z_a = 0$. The balance of concentrations of metabolites in this regime implies that the number of free metabolites in the nucleus, $A_n^f$, passively adjusts to the NC ratio:

$$A_n^f = \frac{1}{1 + \frac{1}{NC}} \cdot A^f \tag{20}$$

As mentioned earlier, the tension of the envelope is responsible for the decrease in the NC ratio. This in turn decreases the number of metabolites inside the nucleus, reinforcing the effect and thus leading to a smaller nuclear volume at saturation (**Figure 4D**). We find the analytical value of the real saturation by using **Equation 19** with $N_n^{tot} = (z_p + 1) \cdot P_n + Q^{eff}$, that is, no metabolites remaining in the nucleus.

Our investigations on the influence of the hydrostatic pressure term in the nested Pump-Leak model lead us to identify another key condition to the nuclear scaling, that is, the presence of folds at the nuclear envelope. Moreover, although not the purpose of this article, using our model to analyze hypo-osmotic shock experiments could allow a precise characterization of the nucleus mechanics.

## Discussion

In this study, we have investigated the emergence of the cell size scaling laws, which are the linear relations between dry mass, nuclear size, and cell size, and which seem ubiquitous in living systems. Using a combination of physical arguments ranging from thermodynamics, statistical physics, polymer physics, mechanics, and electrostatics, we have provided evidence that the robustness of these scaling laws arises from three physical properties: electroneutrality, balance of water chemical potential, and balance of ionic fluxes. The set of associated equations defines a model developed 60 years ago named the Pump-Leak model. The major challenge in probing the origin of the scaling laws using the Pump-Leak model, which we have addressed in this study, is to link a wide range of cell constituents and microscopic biological processes, such as ion transport, translation, transcription, chromatin condensation, nuclear mechanics, to the mesoscopic parameters of the Pump-Leak model (**Table 1**). A host of experimental papers has gathered evidence on these scaling laws and their breakdown over the past century (**Neurohr and Amon, 2020**; **Cantwell and Nurse, 2019b**; **Webster et al., 2009**), but there is still a lack of theoretical understanding of these observations.

In order to go in this direction, we have simplified the Pump-Leak model to its utmost based on the determination of precise orders of magnitude of the relevant parameters. The use of a simplified model focusing on the leading order effects, such as the homeostasis between amino acids and proteins, is a powerful way to isolate and better study the origin of the scaling laws. This is embodied in the accurate predictions, without any adjustable parameters, for the dry mass dilution and the protein dynamics of yeast cells, which are prevented from dividing. A phenomenon that was so far unexplained (**Neurohr et al., 2019**) despite the fact that it is believed to be of fundamental biological importance (**Lengefeld et al., 2021**) by establishing a functional relationship between cell size (and density) and cell senescence, potentially providing a new mechanism driving this important aging process. We emphasize that while we claim that the physical laws and the coarse-grained physical

parameters that constrain cell size are ubiquitous, the specific set of biological processes described in this article at the root of the variation of such parameters is not. In particular, the new biological mechanism, namely the saturation of DNA and mRNA, proposed here to explain dry mass dilution at senescence, is not the only biological mechanism that affects dry mass density. We can indeed quote at least two other identified processes. Variations of the mechanical tension at the plasma membrane during cell spreading were shown to alter the permeability of ion channels, resulting in a volume adaptation at constant dry mass (*Venkova et al., 2022*; *Adar and Safran, 2020*). Similarly, activation of metabolic pathway synthesis such as glycerol for budding yeast cells upon hypertonic stress (*Neurohr and Amon, 2020*), allows us to recover cell volume at almost constant dry mass (since glycerol is a metabolite). Nevertheless, the latter two processes can be easily included in our framework.

The key ingredient of our model is the consideration of small osmolytes and in particular metabolites and small ions. Their high number of fractions among cell-free osmolytes implies that they dominate the control of cell volume. We make three quantitative predictions from this finding. (1) The homeostasis between amino acids and proteins, originating from the enzymatic control of the amino acid pool, explains the dry mass density homeostasis. The disruption of homeostasis, due to mRNA crowding by ribosomes or pharmacological treatment such as rapamycin, is predicted to lead to dry mass dilution upon cell growth due to the saturation of the protein content while the number of amino acids and thus the volume keeps increasing with time. (2) The dry mass dilution observed at mitotic entry for mammalian cells can naturally be explained by the release of counterions condensed on the chromatin, leading to the increase in the number of osmolytes inside the cell and to the subsequent influx of water to ensure osmotic pressure balance at the plasma membrane. (3) The robustness of the NC ratio to the predicted value $\frac{P_n}{P_c}$ is due to the high pool of metabolites within cells, resulting in the dilution of the chromatin (free) counterions that do not scale during growth.

Interestingly, only a few amino acids represent most of the pool of the metabolites possessed by the cell, that is, glutamate, glutamine, and aspartate. This emphasizes their crucial role on cell and nucleus sizes. Our investigations thus link two seemingly distinct hallmark of cancers: the disruption of the cell size scaling laws such as the abnormal karyoplasmic ratio, historically used to diagnose cancer, and metabolic reprogramming, some cancer cells showing an increased consumption of their pool of glutamate and glutamine to fuel the TCA cycle; hence, enhancing their proliferation and invasiveness (*Altman et al., 2016*). This may thus represent possible avenues for future research related to the variability of nucleus size in cancer cells (*Rizzotto and Schirmer, 2017*). Moreover, the large pool of metabolites is a robust feature throughout biology (*Park et al., 2016*), making it one of the main causes of the universality of the cell size scaling laws observed in yeasts, bacteria, and mammalian cells. We believe that the more systematic consideration of such small osmolytes will allow us to understand nontrivial observations. For instance, the recent observation of the increase in diffusivities in the nucleus after blocking nuclear export is explained in our model by the decoupling between protein and amino acid homeostasis after the impingement of ribosome synthesis, a step that requires nuclear export (*Lemière et al., 2022*). Several other published data also seem highly related to the predictions of the model of growth that we propose (*Knapp et al., 2019*; *Odermatt et al., 2021*). We leave open the precise comparison of our model's predictions with these data for future studies.

## The nucleoskeletal theory

To study the nuclear-scaling law, we developed a model for nuclear volume by generalizing the Pump-Leak model that includes both nuclear mechanics, electrostatics, and four different classes of osmolytes. The clear distinction between these classes of components is crucial according to our analysis and is new. (1) Chromatin, considered as a polyelectrolyte gel, does not play a direct role in the osmotic pressure balance because its translational entropy is vanishingly small. Yet, it plays an indirect role on nuclear volume through its counterions. This creates an asymmetry in our system of equations, leading to the unbalance of ionic concentrations across the nuclear envelope and to the appearance of a nuclear envelope potential related to the density of chromatin. (2) Proteins are considered trapped in the nucleus, their number being actively regulated by nucleo-cytoplasmic transport. (3) Metabolites are considered freely diffusable osmolytes through the nuclear envelope but not through the plasma membrane. Note that proteins that have a mass smaller than the critical value 30–60 kDa (*Milo and Phillips, 2015*) are not trapped in the nucleus as they can freely diffuse throughout the nuclear pores. This nevertheless represents more a semantic issue than a physical one, and permeant

proteins are rigorously taken into account as metabolites in the model, but are negligible in practice due to the larger pool of metabolites. (4) Free ions are able to diffuse through the plasma membrane and the nuclear envelope.

As a consequence, we show that the nuclear scaling originates from two features. The first one is the balance of osmotic pressures at the nuclear envelope that we interpret as the result of the nonlinear elastic properties of the nucleus likely due to the presence of folds in the nuclear membrane of mammalian cells. Interestingly, yeast cells do not possess lamina such that the presence of nuclear folds may not be required for the scaling. In this regard, our model adds to a recently growing body of evidence suggesting that the osmotic pressure is balanced at the nuclear envelope in isotonic conditions (*Deviri and Safran, 2022*; *Lemière et al., 2022*; *Finan et al., 2009*). The second feature is the presence of the large pool of metabolites accounting for most of the volume of the nucleus. This explains why nuclear scaling happens during growth while the number of chromatin counterions does not grow with cell size.

Interestingly, although not the direct purpose of this article, our model offers a natural theoretical framework to shed light on the debated nucleoskeletal theory (*Webster et al., 2009*; *Cantwell and Nurse, 2019b*). Our results indicate that the genome size directly impacts the nuclear volume only if the number of (free) counterions of chromatin dominates the number of trapped proteins and the number of metabolites inside the nucleus. We estimate that this number is comparable to the number of trapped proteins while it is about 60 times smaller than the number of metabolites. This is in agreement with recent observations that genome content does not directly determine nuclear volume (*Cantwell and Nurse, 2019b*). Although not directly, chromatin content still influences nuclear volume. Indeed, nuclear volume (*Equation A.61*) is mainly accounted for by the number of metabolites, which passively adjusts according to (Equation 55) ; *Jevtić and Levy, 2014*. In the simple case, of diluted chromatin and no NE potential, metabolite concentration is balanced and $NC = \frac{P_n}{P_c}$, such that the metabolite number depends on two factors (*Equation 20*). The first one is the partitioning of proteins, $\frac{P_n}{P_c}$, that is biologically ruled by nucleo-cytoplasmic transport in agreement with experiments that suggest that the nucleo-cytoplasmic transport is essential to the homeostasis of the NC ratio (*Cantwell and Nurse, 2019b*). The second one is the total number of metabolites, ruled by the metabolism *Equation 9*, which ultimately depends upon gene expression (Appendix 1, section 4.1). This prediction is in line with genetic screen experiments done on fission yeast mutants (*Cantwell and Nurse, 2019a*). However, when the chromatin charge is not diluted, which is likely to occur for cells exhibiting high nuclear envelope potential such as some cancer cells, our theory predicts that the number of metabolites in the nucleus also directly depends on the chromatin content due to electrostatic effects. This highlights the likely importance of chromatin charge in the nuclear-scaling breakdown in cancer.

## Role of nuclear envelope breakdown in cell volume variations

The nested Pump-Leak model predicts that the cell swells upon NEB if the nuclear envelope is under tension. NEB occurs at prometaphase and does not explain most of the mitotic swelling observed in *Son et al., 2015*; *Zlotek-Zlotkiewicz et al., 2015*, which occurs at prophase. Within our model based on counterion release, mitotic swelling is either associated with cytoplasm swelling if the released counterions leave the nucleus or with nuclear swelling if they remain inside. In the latter case, swelling at prophase would be hindered by an increase in nuclear envelope tension, and additional swelling would occur at NEB. This prediction can be tested by artificially increasing the nuclear envelope tension through strong uniaxial cell confinement (*Berre et al., 2014*), which would synchronize mitotic swelling with NEB.

## Physical grounds of the model

Physically, why can such a wide range of biological phenomena be explained by such a simple theory? A first approximation is that we calculated the osmotic pressure considering that both the cytoplasm and the nucleus are ideal solutions. However, it is known that the cytoplasm and the nucleoplasm are crowded (*Feig et al., 2015*; *McGuffee and Elcock, 2010*). The qualitative answer again comes from the fact that small osmolytes constitute the major part of the free osmolytes in a cell so that steric and short-range attractive interactions are only a small correction to the osmotic pressure. We confirm this point by estimating the second virial coefficient that gives a contribution to the osmotic pressure

only of order 2 kPa (see Appendix 1, section 3.1), typically two orders of magnitude smaller than the ideal solution terms (*Table 1*). However, note that we still effectively take into account excluded volume interactions in our theory through the dry volume $R$. Moreover, we show in Appendix 1, section 7.1 that although we use an ideal gas law for the osmotic pressure, the Donnan equilibrium effectively accounts for the electrostatic interactions. Finally, our theory can be generalized to take into account any ions species and ion transport law while keeping the same functional form for the expressions of the volume (*Equation A.21*) as long as only monovalent ions are considered. This is a very robust approximation because multivalent ions such as calcium are in the micromolar range. Together, these observations confirm that the minimal formulation of the Pump-Leak model that we purposely designed is well adapted to study cell size.

## Future extensions of the theory

As a logical extension of our results, we suggest that our framework be used to explain the scaling of other membrane bound organelles such as vacuoles and mitochondria (*Chan and Marshall, 2010*). Provided that the organelles are constrained by the same physical laws, namely the balance of water chemical potential, the balance of ionic fluxes and the electroneutrality condition, we show in Appendix 1 (*Equation A.83*) that the incorporation of other organelles into our framework leads to the same equations as for the nucleus, thus pointing out that the origin of the scaling of other organelles may also arise from the balance of osmotic pressures. The precise experimental verification of such prediction for a specific organelle is left opened for future studies. We also propose that our theory be used to explain the scaling of membraneless organelles such as nucleoids (*Gray et al., 2019*). Indeed, the Donnan picture that we are using does not require membranes (*Barrat and Joanny, 1996*). However, we would have to add other physical effects in order to explain the partitioning of proteins between the nucleoid and the bacterioplasm.

Taken as a whole, our study demonstrates that cell size scaling laws can be understood and predicted quantitatively on the basis of a remarkably simple set of physical laws ruling cell size as well as a simple set of universal biological features. The new interpretations of previous empirical biological phenomena that our approach allows to provide indicates that this theoretical framework is fundamental to cell biology and will likely benefit the large community of biologists working on cell size and growth.

## Acknowledgements

We thank Matthieu Piel and the members of his team, in particular Damien Cuvelier, Alice Williart, Guilherme Nader, and Nishit Srivastava, for insightful discussions and for showing us data that originally motivated the theory on nuclear volume; Thomas Lecuit for introducing us to the cell size scaling laws with his 2020 course at College de France entitled 'Volume cellulaire determinants Physico-chimiques et regulation'; Pierre Recho for showing us his seminal work on nuclear volume; the members of the UMR 168, Amit Singh Vishen, Ander Movilla, Sam Bell, Mathieu Dedenon, and Joanna Podkalicka, as well as Dan Deviri for fruitful discussions. The Sens laboratory is a member of the Cell(n)Scale Labex.

## Additional information

### Competing interests

Pierre Sens: Reviewing editor, *eLife*. The other authors declare that no competing interests exist.

### Funding

| Funder | Grant reference number | Author |
| --- | --- | --- |
| Programme d'investissements d'avenir | ANR-11-LABX-0038 | Romain Rollin |
| Programme d'investissements d'avenir | ANR-10-IDEX-0001-02 | Romain Rollin |

| Funder | Grant reference number | Author |
|--------|----------------------|--------|

The funders had no role in study design, data collection and interpretation, or the decision to submit the work for publication.

## Author contributions

Romain Rollin, Conceptualization, Resources, Software, Formal analysis, Investigation, Visualization, Methodology, Writing - original draft, Project administration; Jean-François Joanny, Pierre Sens, Conceptualization, Supervision, Validation, Methodology, Writing – review and editing

## Author ORCIDs

Romain Rollin [ID] http://orcid.org/0000-0002-6042-234X
Jean-François Joanny [ID] http://orcid.org/0000-0001-6966-3222
Pierre Sens [ID] http://orcid.org/0000-0003-4523-3791

## Decision letter and Author response

Decision letter https://doi.org/10.7554/eLife.82490.sa1
Author response https://doi.org/10.7554/eLife.82490.sa2

# Additional files

## Supplementary files

• Transparent reporting form

## Data availability

All data analysed during this study are included in the manuscript and supporting file; Source Data files have been provided for Figures 2 and 4.Figure 2 - Source Data 1 to 4 contain the experimental data used to fit and validate our theory in the panels B to E of Figure 2. These data are extracted from *Neurohr et al., 2019*.Figure 4 - Source Data 1 contains the experimental data used to fit and validate our theory in the panel D of Figure 4. These data are extracted from *Finan et al., 2009*.

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

## Appendix 1

### 1.1 Pump-Leak model fundamental equations

In this section, we derive and discuss the three physical constraints that we used throughout our article to study cell volume regulation. These results are classical and can be found in a reference textbook such as *Sten-Knudsen, 2007*.

### 1.1.1 Electroneutrality

The intrinsic length scale associated to the Poisson equation is the Debye length. It appears explicitly in the linearized version of the Poisson equation also called the Debye–Huckel equation. It reads

$$\lambda_D = \left( \frac{1}{4\pi \cdot l_b \cdot (n^+ + n^-)} \right)^{\frac{1}{2}} \tag{A.1}$$

where $l_b = \frac{e^2}{4\pi k_B T \epsilon_r \epsilon_0}$ is the Bjerrum length, which qualitatively corresponds to the distance between two elementary charges at which the electrostatic energy will be comparable to the thermal energy. $l_b \approx 0.7\,\text{nm}$ in water at $300\,\text{K}$. In the unit used in this article (concentrations in mmol), the Debye length can be estimated using the following formula:

$$\lambda_D \approx \frac{9.7}{\sqrt{n^+(mM) + n^-(mM)}} \cdot nm \tag{A.2}$$

For a typical mammalian cell $n^+(\text{mM}) + n^-(\text{mM}) \approx 180\,\text{mM}$ (*Table 1*) which leads to a Debye length $\lambda_D \approx 0.7\,\text{nm}$. Thus, the Debye length is at least three orders of magnitude smaller than the typical radius of a cell or of a nucleus. This justifies the approximation of electroneutrality used throughout the main article for length scales much larger than the Debye length.

### 1.1.2 Balance of water chemical potential

We define the osmotic pressure as

$$\Pi = -\frac{1}{v_w} \cdot (\mu_w - \mu_w^*) \tag{A.3}$$

where $v_w$ is the molecular volume of water, and $\mu_w$, $\mu_w^*$ are the chemical potential of water, respectively, in the real solution and in a pure water solution. Assuming that water is incompressible, the chemical potential of pure water reads:

$$\mu_w^* = \mu_0(T) + v_w \cdot P \tag{A.4}$$

where $P$ is the hydrostatic pressure, and $\mu_0(T)$ the standard chemical potential of pure water that only depends on the temperature. Injecting *Equation A.4* into *Equation A.3*, we express the chemical potential of water as

$$\mu_w = \mu_0(T) + v_w \cdot (P - \Pi) \tag{A.5}$$

As emphasized in the main text, we assume that water equilibrates instantaneously at the timescale of cellular growth. As such, we impose that the chemical potential of water is balanced on each side of the plasma membrane: $\mu_w^c = \mu_w^o$, where the upper-script $o$ and $c$ refer to the outer medium and the cellular medium. Using the expression of the chemical potential of water derived in *Equation A.5* in each medium, we obtain

$$P^c - P^0 = \Pi^c - \Pi^0 \tag{A.6}$$

The latter equation corresponds to the first equality in *Equation 2*. Note that it does not assume the dilute solution assumption. To further express the osmotic pressure, we assume that the solutions, either the cellular or the outer media, are dilute and hence we use a perfect gas law for the osmotic pressure. This is how we obtained the second equality in *Equation 2*.

### 1.1.3 Balance of ionic fluxes

The total flux $J$ of cations – respectively anions – is decomposed between three main contributions: active pumping, electrical conduction, and entropic diffusion. For simplicity, we assumed that only

cations are pumped out of the cell. This simplifying choice was made to model the Na/K pump, which is one of the most relevant cationic pumps. Though, we show in Appendix 1, section 2.1.3 that this assumption is not critical since the equations keep the same functional form if it is relaxed. We also assume a slight deviation from equilibrium so that the ion fluxes are proportional to the chemical potential difference between both sides of the plasma membrane. As a convention, we choose $J$ to be positive when ions are entering the cell. At steady state, the fluxes vanish:

$$\begin{cases} J_{+,tot} = g^+ \cdot \left[ -e \cdot U_c - k_B T \left( \frac{n^+}{n_0} \right) \right] - p = 0 \\ J_{-,tot} = g^- \cdot \left[ e \cdot U_c - k_B T \cdot ln \left( \frac{n^-}{n_0} \right) \right] = 0 \end{cases} \tag{A.7}$$

where $p$ is the pumping flux, $g^{\pm}$ are the membrane conductivities for cations and anions, and $U_c$ is the cell transmembrane potential, which can be expressed as

$$U_c = -\frac{k_B T}{e} \cdot ln \left( \frac{n^+}{n_0} \right) - \frac{p}{g_+} = \frac{k_B T}{e} \cdot ln \left( \frac{n^-}{n_0} \right) \tag{A.8}$$

For **Equation A.8** to be verified, the following relationship between $n^+$ and $n^-$ must be imposed:

$$\begin{cases} n^+ \cdot n^- = \alpha_0 \cdot n_0^2 \\ \alpha_0 = e^{-\frac{p}{k_B T g^+}} \end{cases} \tag{A.9}$$

The latter equation takes the form of a generalized Donnan ratio that includes the active pumping of cations. The usual Donan ratio (**Sten-Knudsen, 2007**) is recovered when $p = 0$. The generalized Donnan ratio **Equation 3**, together with the electroneutrality condition **Equation 1**, yields analytic expressions for the ionic densities $n^+$ and $n^-$ (the notations are defined in the main text):

$$\begin{cases} n^+ = \frac{zx + \sqrt{(zx)^2 + 4\alpha_0 n_0^2}}{2} \\ n^- = \frac{-zx + \sqrt{(zx)^2 + 4\alpha_0 n_0^2}}{2} \end{cases} \tag{A.10}$$

The cell osmotic pressure can thus be expressed as:

$$\frac{\Pi}{k_B T} = \sqrt{(zx)^2 + 4\alpha_0 n_0^2} + x \tag{A.11}$$

## 2.1 General expressions of the volume in the Pump-Leak model

The system of equation formed by **Equations 1–3** is nonlinear and cannot be solved analytically in its full generality. One complexity arises from the difference of hydrostatic pressure $\Delta P$. Intuitively, if the volume increases, the surface increases, which may in some situations increase the tension of the envelope and in turn impede the volume growth. Mathematically, Laplace law relates the difference of hydrostatic pressure to the tension $\gamma$ and the mean curvature of the interface, which simplifies to the radius of the cell $R$ in a spherical geometry. The difference of hydrostatic pressure then reads:

$$\Delta P = \frac{2\gamma}{R} \tag{A.12}$$

In the case where the interface exhibits a constitutive law that is elastic $\gamma = K \cdot \frac{S - S_0}{S_0}$, it is easy to see that $\Delta P$ exhibits power of $V$, which makes the problem nonanalytical. However, we can get around this limitation in two biologically relevant situations:

- When $\Delta P$ is negligible. As shown in **Table 1**, this happens for mammalian cells that do not possess cellular walls.
- When $\Delta P$ is buffered by biological processes. We argue that this situation applies for yeasts and bacteria during growth. Indeed, if the volume increase is sufficiently slow, one can hypothesize that cells have time to add materials to their cellular walls such that the tension does not increase during growth.

We give the corresponding analytical expressions under these two hypotheses in the next two sections.

### 2.1.1 Analytical expression of the volume when hydrostatic pressure difference is negligible

The balance of water chemical potential (*Equation 2*), neglecting the difference of pressure and injecting the expressions for the ionic densities (*Equation A.10*), leads to the following equation for the density of impermeant molecules $x$:

$$(z^2 - 1) \cdot x^2 + 4n_0 \cdot x - 4 \cdot (1 - \alpha_0) \cdot n_0^2 = 0 \tag{A.13}$$

Solving this equation and using the definition of the density of impermeant molecules $x = \frac{X}{V-R}$ yield the expression for the volume of the cytoplasm:

$$\begin{cases} V - R = \frac{k_B T \cdot N^{tot}}{\Pi_0} \\ N^{tot} = X \cdot \frac{(z^2 - 1)}{-1 + \sqrt{1 + (1 - \alpha_0)(z^2 - 1)}} \end{cases} \tag{A.14}$$

The volume can thus be written as an ideal gas law with a total number of free osmolytes $N^{tot}$. This number takes into account the different ions and is thus larger than the actual number of impermeant molecules $X$. In the limit of very fast pumping – $\alpha_0 \to 0$ – (*Equation A.14*) reduces to the expression given in the main text (*Equation 4*). On the other hand, when $\alpha_0 \to 1$, the volume diverges. This is due to the fact that the osmotic pressure in the cell (*Equation A.11*) is always higher than the external osmotic pressure in this limit. In the absence of a hydrostatic pressure difference, the cell would swell to make the concentration of trapped macromolecules $x$ infinitely small. This shows the importance of ion pumps to achieve a finite volume for mammalian cells.

### 2.1.2 Analytical expression of the volume when $\Delta P$ is buffered

The same procedure can be used when $\Delta P$ is buffered (independent of the volume). The final expression reads

$$\begin{cases} V - R = \frac{k_B T \cdot N^{tot}(\Delta P)}{(\pi_0 + \Delta P)} \\ \\ N^{tot}(\Delta P) = X \cdot \frac{z^2 - 1}{-1 + \sqrt{1 + (z^2 - 1) \cdot \left(1 - \frac{\alpha_0}{\left(1 + \frac{\Delta P}{k_B T \cdot 2n_0}\right)^2}\right)}} \end{cases} \tag{A.15}$$

Interestingly, the wet volume $V - R$ remains proportional to the number of impermeant molecules $X$ in this limit.

### 2.1.3 Analytical expression of the volume for an arbitrary number of ions and active transports

In this subsection, we generalize the Pump-Leak model to any type of ions and any ionic transport. Each ion can be actively transported throughout the membrane. Importantly, we show that – as long as ions are monovalent – the Pump-Leak model equations and solutions take the same functional form as the two-ion model used in the main text. We use the same notations as in the main text (section 'Pump and leak model' and *Table 1*), except that we now add subscript $i$ to refer to the ion of type $i$. For instance, $z_i^-$ – respectively $n_i^-$ – refers to the valency – respectively the concentration – of the anion $i$. The densities of positive/negative charges in the cell read:

$$\begin{cases} d^+ = \sum_j z_j^+ \cdot n_j^+ \\ d^- = \sum_j z_j^- \cdot n_j^- \end{cases} \tag{A.16}$$

Electroneutrality thus simply reads:

$$d^+ - d^- - z \cdot x = 0 \tag{A.17}$$

Balancing ionic fluxes for each ion types, as in *Equation A.7*, leads to:

$$\begin{cases} n_j^{+/-} = n_j^0 \cdot \alpha_j \cdot e^{(-/+)\cdot z_j \cdot \frac{e \cdot U_C}{k_B T}} \\ \alpha_j = e^{-\frac{p_j}{g_j}} \end{cases} \tag{A.18}$$

Using *Equations A.16–A.18* and assuming that all ions are monovalent, the product of the cationic and anionic densities can be expressed as:

$$d^+ \cdot d^- = \underbrace{\left( \sum_j n_j^{+,0} \cdot \alpha_j \right) \cdot \left( \sum_i n_i^{-,0} \cdot \alpha_i \right)}_{\underset{def}{\equiv} \tilde{\alpha}(n_i^0)} \tag{A.19}$$

and the analytical solution of the full problem reads:

$$\begin{cases} d^+ = \frac{zx + \sqrt{(zx)^2 + 4\tilde{\alpha}(n_i^0)}}{2} \\ \\ d^- = \frac{-zx + \sqrt{(zx)^2 + 4\tilde{\alpha}(n_i^0)}}{2} \end{cases} \tag{A.20}$$

$$\begin{cases} V - R = \frac{k_B T \cdot N^{tot}(\Delta P)}{(\pi_0 + \Delta P)} \\ \\ N^{tot}(\Delta P) = X \cdot \dfrac{z^2 - 1}{-1 + \sqrt{1 + (z^2 - 1) \cdot \left( 1 - \dfrac{4\tilde{\alpha}(n_i^0)}{\left( \frac{1}{k_B T} \cdot \Pi_0 + \frac{1}{k_B T} \cdot \Delta P \right)^2} \right)}} \end{cases} \tag{A.21}$$

which shows a similar form as the two-ion model (*Equations A.10; A.15*).

## 3.1 Order of magnitudes

Throughout the main text, we used order of magnitudes to guide our investigations and justify our approximations. For the sake of readability, we gather all the parameter significations, values, and origins in *Appendix 1—table 1*.

**Appendix 1—table 1.** Description and values of the parameters used for the order of magnitudes.

| Symbol | Typical Value | Meaning |
|---|---|---|
| $\rho$ | $0.1\,\text{kg.L}^{-1}$ | Typical dry mass density in a mammalian cell (*Zlotek-Zlotkiewicz et al., 2015*) |
| $\mathcal{M}_a$ | $100\,\text{Da}$ | Average mass of an amino acid (*Milo and Phillips, 2015*) |
| $l_p$ | $400\,\text{a.a}$ | Average length of a eukaryotic protein (*Milo and Phillips, 2015*) |
| $l_{mRNA}$ | $3 \cdot l_p$ | Average length of an mRNA (*Milo and Phillips, 2015*) |
| $l_{bp}$ | $1/3\,\text{nm}$ | Average length of one base pair |
| $Q_{bp}$ | $2$ | Average number of negative charges per base pair |
| $L_{nucleosome}$ | $200\,\text{bp}$ | Average length of DNA per nucleosome |
| $L_{link}$ | $53\,\text{bp}$ | Length of the DNA linking two histones |
| $L_{wrap}$ | $147\,\text{bp}$ | Length of the DNA wrapped around one histone |

*Appendix 1—table 1 Continued on next page*

*Appendix 1—table 1 Continued*

| Symbol | Typical Value | Meaning |
|---|---|---|
| $u_{DNA}$ | 4 | Manning parameter for pure DNA, i.e., 75% of the charges will be screened by manning condensation |
| $L_{tot}$ | $6 \cdot 10^9$ bp | Total length of the DNA within a diploid human cell |
| $Q_{hist}$ | 76 | Average number of positive charges per histone at less than 1 nm from the wrapped DNA backbone (*Materese et al., 2009*) |
| $Q_{wrap}$ | 174 | Average number of condensed counterions around the wrapped DNA (*Materese et al., 2009*) |
| $l_b$ | 0.7 nm | Bjerrum length in water at 300k |
| $K$ | 25 mN/m | Stretching modulus of lamina (*Dahl et al., 2004*) |

### 3.1.1 Protein concentration

We use data published in *Zlotek-Zlotkiewicz et al., 2015* to estimate the typical concentration of proteins in mammalian cell $p_{tot}$ as

$$p_{tot} = \%_p^{mass} \cdot \frac{\rho}{\mathcal{M}_a \cdot l_p \cdot (1 - \frac{R}{V})} \sim 2 \, \text{mMol} \tag{A.22}$$

where $\%_p^{mass}$ is the fraction of dry mass occupied by proteins (*Figure 1C*).

### 3.1.2 mRNA to protein fraction

In *Figure 1B*, we neglected the contribution of mRNAs to the wet volume of the cell. The rationale behind this choice is twofold. (1) Proteins represent less than 1% of the wet volume. (2) The mRNA to protein number fraction is estimated to be small due to the fact that the mass of one mRNA is nine times greater than the one of a protein while the measured fraction of mRNA to dry mass is of the order 1% (*Milo and Phillips, 2015*):

$$\frac{M_{tot}}{P_{tot}} = \frac{\mathcal{M}_p}{\mathcal{M}_{mRNA}} \cdot \frac{mass_{mRNA}}{mass_p} \sim \frac{1}{500} \tag{A.23}$$

Thus, mRNAs contribute even less than proteins to the wet volume.

### 3.1.3 Metabolite concentration

We find the metabolite concentration self-consistently by enforcing balance of osmotic pressure at the plasma membrane (*Equation 2*):

$$a^f = 2n_0 - p_{tot} - n^+ - n^- \sim 118 mMol \tag{A.24}$$

where the concentrations of ions were reported in *Milo and Phillips, 2015* (see *Table 1*). This high value of metabolite concentration is coherent with reported measurements (*Park et al., 2016*).

### 3.1.4 Contribution of osmolytes to the wet volume of the cell (*Figure 1B*)

The contribution of osmolytes to the wet volume fraction is simply equal to the ratio of the osmolyte concentration to the total external ionic concentration, here equal to $2n_0$ (*Equation 4*). The concentration of specific amino acids and metabolites was estimated using their measured proportion in the metabolite pool (*Park et al., 2016*) times the total concentration of metabolites $a^f$ (*Equation A.24*).

### 3.1.5 Amino acids contribution to the dry mass

One of the main conclusions from our order of magnitude estimates is that amino acids play an essential role in controlling the volume but have a negligible contribution to the cell's dry mass. This originates

from the large average size of proteins $l_p \sim 400 a.a$. The contribution of amino acids to the dry mass reads:

$$\%_{a.a}^{mass} = \%_{a.a}^{number} \cdot \frac{a^f}{P_{tot}} \cdot \frac{\%_p^{mass}}{l_p} \sim 6\% \tag{A.25}$$

where $\%_{a.a}^{number} \sim 73\%$ is the number of fraction of amino acids among metabolites.

### 3.1.6 Effective charge of chromatin
The average effective charge per nucleosome is estimated to be:

$$Q_{pernucleosome}^{eff} = L_{Link} \cdot \frac{Q_{bp}}{u_{DNA}} + L_{wrap} \cdot Q_{bp} - Q_{hist} - Q_{wrap} = 71 \tag{A.26}$$

where the right-hand side can be understood as the total negative charge of pure DNA, screened in part by histone positive charges and by the manning condensed counterions. Note that the number of condensed counterions $Q_{wrap}$ around the wrapped DNA simulated in *Materese et al., 2009* is similar to the value expected by the manning theory that we estimate to be 164 elementary charges.

The number of nucleosomes is simply $N_{hist} = \frac{L_{tot}}{L_{nucleosomes}} = 3 \cdot 10^7$ such that the effective charge of chromatin is estimated to be:

$$Q^{eff} = 2 \cdot 10^9 \tag{A.27}$$

Note that the size of the budding yeast genome is 12 Mbp (*Goffeau et al., 1996*) such that, assuming for simplicity that 75% of this charge is screened either by manning condensation or by histones, we find that $Q^{eff} \sim 6 \cdot 10^6$.

### 3.1.7 Condensed counterions on chromatin
The condensed counterions on chromatin can simply be found from the effective charge of the chromatin, the total charge of pure DNA, and the charge of histones. We obtain:

$$Q^{cond} = Q_{bp} \cdot L_{tot} - Q^{eff} - Q_{hist} \cdot N_{hist} \sim 8 \cdot 10^9 \tag{A.28}$$

### 3.1.8 Estimation of the amplitude of the mitotic swelling
At mitosis, cells have doubled their genome content such that we double the number of condensed counterions estimated earlier for a diploid mammalian cell. Using the Pump-Leak model, we compute the amplitude of swelling if all the chromatin condensed counterions were released at the same time, assuming an external osmolarity of $n_0 = 100 - 150\,\text{mM}$:

$$\Delta V = \frac{2 \cdot Q^{cond}}{2 n_0} \sim 100 \mu \text{m}^3 \tag{A.29}$$

Note that $\Delta V$ must scale with the number of genome duplications. For instance, for tetraploid cells, the previous amplitude must be doubled.

### 3.1.9 Average charge of proteins and metabolites
The average charge of proteins used in the article is $z_p \sim 0.8$. To obtain this value, we used the data analysis performed in *Requião et al., 2017*. The authors plotted the net-charge frequency histogram of the amino acid segments in all 551,705 proteins from the SwissProt database respectively assuming that the histidine charge is 0 or 1. To obtain $z_p$, we computed the average of the histogram assuming neutral histidines. This choice is reasonable because the Pka of the histidine amino acid is of order 6 while the typical physiological pH is of order 7.4, so that only 4% of histidines are charged in the cell. Note that this estimate is not very precise since it does not take into account the relative proportion of proteins. We nevertheless checked that our conclusions remain valid by varying $z_p$ from 0 to 3.

The average charge of metabolites is assumed to be $z_a \sim 1$ since glutamate is the most abundant (*Park et al., 2016*). We have checked that changing this parameter does not alter our conclusions.

### 3.1.10 Absolute number of osmolytes

To obtain the **Figure 4**, we had to estimate the parameter $NC_1 = \frac{P_n}{P_c}$ and thus, the number of protein trapped inside the nucleus $P_n$ at the beginning of interphase. The total number of proteins and metabolites at the beginning of interphase is simply obtained by multiplying the concentration of proteins $p_{tot}$, $m_{tot}$ estimated earlier by the volume at the beginning of interphase, measured to be equal to $1250\mu m^3$(**Zlotek-Zlotkiewicz et al., 2015**), minus the dry volume that roughly represent 30% of the total volume (**Table 1**).

$$P_{tot} = p_{tot} \cdot (V - R) \sim 10^9 \qquad (A.30)$$

$$A^f = \frac{a^f}{p_{tot}} \cdot P_{tot} \sim 60 \cdot 10^9 \qquad (A.31)$$

In the regime where the chromatin is diluted (large amount of metabolites), the NC ratio can be well approximated by $NC_1 = \frac{P_n}{P_c}$. Usual values of NC reported in the literature typically range from 0.3 to 0.6 (**Guo et al., 2017**; **Wu et al., 2022**). We thus estimate reasonable values of $P_n$ as

$$P_n = \frac{NC}{1+NC} \cdot P_{tot} \sim 3 \cdot 10^8 \qquad (A.32)$$

Note that we also used the numerical solutions of **Equation A.55** in Appendix 1, section 6.1 to infer $NC_1$ from $NC$ exactly. This method made no qualitative difference to the results plotted in **Figure 4**.

For budding yeasts, similarly to the chromatin effective charge, the number of protein in the nucleus is typically three orders of magnitudes smaller. Indeed, it is reported that $P_{tot} \sim 5 \cdot 10^7$ both in **Ho et al., 2018**; **Ghaemmaghami et al., 2003**; **von der Haar, 2008**. Moreover, using a typical NC ratio of 0.1 (**Lemière et al., 2022** and **Equation A.32**), we find that $P_n \sim 4 \cdot 10^6$.

### 3.1.11 Estimation of an upper bound for the hydrostatic pressure difference of the nucleus

Even though the stiffness of the lamina layer is susceptible to vary according to the tissue the cell is belonging to **Swift et al., 2013**, its stretching modulus was reported to range from 1 to 25 mN/m (**Dahl et al., 2004**; **Stephens et al., 2017**). Also, lamina turnover rate is much slower than the actin turnover rate. Together, this suggests that lamina – at the difference to the cortical actin – can sustain bigger pressure difference on longer timescales. This solid-like behavior of lamina was observed during micropipette aspiration of oocyte nuclei through the formation of membrane wrinkles at the pipette entrance (**Dahl et al., 2004**). We thus chose to mathematically model lamina with an elastic constitutive equation when it is tensed (**Equation A.74**). Using Laplace law, we estimate an upper bound for $\Delta P_n$, assuming a typical nuclear radius of 5 μm, to be

$$\Delta P_n \sim \frac{2K}{R} \sim 10^4 Pa \qquad (A.33)$$

### 3.1.12 Estimation of the second virial term in the osmotic pressure

We estimate the steric term in the osmotic pressure to be

$$\Pi_{steric} \sim k_B T \cdot v_p \cdot p_{tot}^2 \sim 2kPa \qquad (A.34)$$

where $v_p$ is the excluded volume per protein, estimated to be $v_p \sim \frac{R}{P_{tot}} \sim 375nm^3$. This corresponds to a protein radius of 4.5 nm, a value coherent with observations (**Milo and Phillips, 2015**). This steric contribution in the osmotic pressure may thus be safely neglected, as $\Pi_{steric} << \Pi_0$.

## 4.1 A cell growth model

We summarize here the equations derived and discussed in the main text (**Equations 6 and 7**). The rates of production of mRNAs and proteins in the nonsaturated and saturated regimes read:

$$\begin{cases} \dot{M}_j = k_0 \cdot \phi_j \cdot P_p - \frac{M_j}{\tau_m}, & if \quad P_p \leq P_p^* \\ \dot{M}_j = k_0 \cdot g_j \cdot \mathcal{N}_p^{max} - \frac{M_j}{\tau_m}, & if \quad P_p \geq P_p^* \end{cases} \qquad (A.35)$$

$$\begin{cases} \dot{P}_j = k_t \cdot \frac{M_j}{\sum M_j} \cdot P_r - \frac{P_j}{\tau_p}, & \text{if} \quad P_r \leq P_r^* \\ \dot{P}_j = k_t \cdot M_j \cdot \mathcal{N}_r^{max} - \frac{P_j}{\tau_p}, & \text{if} \quad P_r \geq P_r^* \end{cases} \tag{A.36}$$

The cut-off values – $P_p^*$, $P_r^*$ – above which the substrates become saturated are obtained by imposing continuities of the production rates at the transition:

$$\begin{cases} P_p^* = \mathcal{N}_p^{max} \cdot \sum g_j \\ P_r^* = \mathcal{N}_r^{max} \cdot \sum M_j \end{cases} \tag{A.37}$$

### 4.1.1 Neither DNA nor mRNAs are saturated: $P_p \leq P_p^*$ and $P_r \leq P_r^*$

The fast degradation rate of mRNAs ensures that their number is determined by the steady state of **Equation A.35**. **Equation 8** with **Equation 7** thus yields an exponential growth of the number of ribosomes $P_r = P_{r,0} \cdot e^{k_r \cdot t}$ (with, $k_r = k_t \cdot \phi_r - 1/\tau_p$) and of any other protein, $P_j = \frac{\phi_j}{\phi_r} \cdot P_r$; where we call $P_{r,0}$ the initial number of ribosomes and we neglect the initial conditions on proteins other than ribosomes due to the exponential nature of the growth. Incorporating the dynamics of growth of the enzyme catalyzing the amino acid biosynthesis $P_e$ into **Equation 9**, we obtain the number of free amino acids in the cell:

$$A^f = \left( \phi_e \cdot \frac{k_{cat}}{k_r} - l_p \right) \cdot \frac{P_r}{\phi_r} \tag{A.38}$$

Using the expression of the volume (**Equation 4**) derived from the Pump-Leak model coupled to our quantitative order of magnitudes, it is straightforward to show that the volume grows exponentially:

$$V = \left( v_p + \frac{(z_{A,f}+1) \cdot \left( \phi_e \cdot \frac{k_{cat}}{k_r} - l_p \right)}{2n_0} \right) \cdot \frac{P_r}{\phi_r} \tag{A.39}$$

where we assumed the dry volume to be mainly accounted for by proteins. Incorporating the previous expressions in the equation for the dry mass density (**Equation 5**), we obtain the homeostatic dry mass density written in the main text (**Equation 10**). These expressions were obtained assuming that neither the DNA nor the mRNA were saturated. Importantly, mRNAs cannot be saturated if DNA is not saturated because the cut-off value $P_r^*$ for which ribosomes saturates mRNAs grows at the same speed as the number of ribosomes: $P_r^* = \mathcal{N}_r^{max} \cdot k_0 \cdot \tau_m \cdot \frac{\phi_p}{\phi_r} \cdot P_r$. Hence, DNA will saturate before mRNAs during interphase, at a time $t^*$ given by:

$$t^* = \frac{1}{k_r} \cdot ln \left( \frac{g_r}{g_p} \cdot \frac{\mathcal{N}_p^{max} \cdot \sum g_j}{P_{r,0}} \right) \tag{A.40}$$

### 4.1.2 DNA is saturated but not mRNAs: $P_p \geq P_p^*$ and $P_r \leq P_r^*$

The only difference with the previous regime is that mRNA number saturates to the value $M_j = k_0 \cdot g_j \cdot \tau_m \cdot \mathcal{N}_p^{max}$. Hence, the threshold $P_r^*$ will saturate to the value:

$$P_r^* = \mathcal{N}_r^{max} \cdot \mathcal{N}_p^{max} \cdot k_0 \cdot \tau_m \cdot \sum g_j \tag{A.41}$$

This allows for the subsequent saturation of mRNAs by ribosomes after a time $t^{**}$, whose expression can be derived after simple algebra as:

$$t^{**} = t^* + \frac{1}{k_r} \cdot ln \left( \frac{g_p}{g_r} \cdot \mathcal{N}_r^{max} \cdot k_0 \cdot \tau_m \right) \tag{A.42}$$

However, before reaching this time, there will not be any consequence on the proteomic dynamics, which still scales with the number of ribosomes $P_j = \frac{\phi_j}{\phi_r} \cdot P_r$. This regime thus still corresponds to an exponential growth and the dry mass density remains at its homeostatic value (**Equation 10**).

### 4.1.3 Both DNA and mRNAs are saturated: $P_p \geq P_p^*$ and $P_r \geq P_r^*$

The dynamics of growth is profoundly impacted by mRNA saturation. The protein number no longer grows exponentially, but saturates to the stationary value $P_j^{stat} = k_t \cdot k_0 \cdot \tau_p \cdot \tau_m \cdot \mathcal{N}_r^{max} \cdot \mathcal{N}_p^{max} \cdot g_j$ after a typical time $t^{**} + \tau_p$ according to:

$$P_j = P_j^{stat} + \left(P_j(t^{**}) - P_j^{stat}\right) \cdot e^{-\frac{t-t^{**}}{\tau_p}} \qquad (A.43)$$

The loss of the exponential scaling of proteins implies a breakdown of the proportionality between amino acid and protein numbers as predicted by the amino acid biosynthesis equation (**Equation 9**). The total amino acid pool in the cell $A_{tot} = A_f + l_p \cdot P_{tot}$ now scales as:

$$A_{tot} = A_{tot}(t^{**}) + k_{cat} \cdot \left[P_e^{stat} \cdot \left(t - t^{**}\right) - \tau_p \cdot \left(P_e(t^{**}) - P_e^{stat}\right) \cdot \left(e^{-(t-t^{**})/\tau_p} - 1\right)\right] \qquad (A.44)$$

with $A_{tot}(t^{**}) = \frac{\phi_e}{\phi_r} \cdot \frac{k_{cat}}{k_r} \cdot \mathcal{N}_r^{max} \cdot \mathcal{N}_p^{max} \cdot k_0 \cdot \tau_m \cdot \sum g_j$. Although expressions still remain analytical in the transient regime and were implemented in **Figure 2** in order to quantitatively test our theory, we avoid analytical complications here by writing expressions after saturation has been reached, that is, after a typical time $t^{**} + \tau_p$. The volume thus increases linearly with time:

$$V^{lin} = v_p \cdot P_{tot}^{stat} + \frac{(z_{Af}+1) \cdot \left(A_{tot}(t^{**}) + k_{cat} \cdot P_e^{stat} \cdot (t-t^{**}) - l_p \cdot P_{tot}^{stat}\right)}{2n_0} \qquad (A.45)$$

As emphasized in the main text, the fundamental property of this regime is that the dry mass density is predicted to decrease with time with no other mechanism than a simple crowding effect on mRNAs (see **Equation 11** in the main text).

## 4.1.4 Quantification of the model of growth with published data

Many of the parameters involved in the growth model can be obtained independently, so that four parameters suffice to fully determine the volume, the amount of protein, and the dry mass density during interphase growth. Here, we summarize the equations used to fit the data displayed in **Figure 2**. The volume can be expressed as:

$$V = \begin{cases} v_1 \cdot e^{k_r \cdot t}, & \text{if } t \leq t^{**} \\ v_2 \cdot (t - t^{**}) + v_3 \cdot e^{-(t-t^{**})/\tau_p} + v_4, & \text{if } t \geq t^{**} \end{cases} \qquad (A.46)$$

in which $(v_1, v_2, v_3, v_4)$ are volumes that can be, if needed, expressed function of the previously defined parameters. We obtain $(v_2, v_3, v_4)$ as a function of $v_1$, $\tau_p$ and $t^{**}$ by imposing regularity constraints on the volume and growth rate:

$$\begin{cases} v_3 = \tau_p^2 \cdot k_r^2 \cdot v_1 \cdot e^{k_r \cdot t^{**}} \\ v_2 = \frac{v_3}{\tau_p} + k_r \cdot v_1 \cdot e^{k_r \cdot t^{**}} \\ v_4 = v_1 \cdot e^{k_r \cdot t^{**}} - v_3 \end{cases} \qquad (A.47)$$

Similarly, the normalized total number of protein can be expressed as:

$$P_{tot}/P_{tot}(1h) = \begin{cases} e^{k_r \cdot (t-1)}, & \text{if } t \leq t^{**} \\ p_1 + p_2 \cdot e^{-(t-t^{**})/\tau_p}, & \text{if } t \geq t^{**} \end{cases} \qquad (A.48)$$

Again imposing regularity constraints at the mRNA saturating transition allows us to relate $p_1$ and $p_2$ to $k_r$, $\tau_p$ and $t^{**}$.

$$\begin{cases} p_2 = -\tau_p \cdot k_r \cdot e^{k_r \cdot (t^{**}-1)} \\ p_1 = e^{k_r \cdot (t^{**}-1)} - p_2 \end{cases} \qquad (A.49)$$

Finally, we can express the total mass density $\rho_{tot}$ of the cell (including water and dry mass, see **Figure 2**) using the expressions of total protein number and volume **Equations A.46; A.48**:

$$(\rho_{tot} - \rho_w)/(\rho_{tot,0} - \rho_w) = \begin{cases} 1, & \text{if } t \leq t^{**} \\ \frac{P_{tot}}{P_{tot}(0h)} \cdot \frac{v_1}{V}, & \text{if } t \geq t^{**} \end{cases} \qquad (A.50)$$

We use a density of water 4% larger than that of pure water ($\rho_{w,eff} = 1.04$kg/L instead of $\sim 1$ kg/L) to compensate for our approximation to consider the dry mass as entirely made of proteins. Proteins are known to only occupy $\%_p^{mass} = 0.6$ of the dry mass, itself being of order $\rho = 0.1$ kg/L (*Appendix 1—table 1*). Thus, we simply use as the effective water mass density, $\rho_{w,eff} = \rho_w + (1 - \%_p^{mass}) \cdot \rho \sim 1.04$ kg/L.

### 4.1.5 Fitting procedure

We detail in this appendix the method used to determine the four fitting parameters: $\tau_p, t^{**}, k_r, v_1$ from the cell volume data (*Figure 2B*). Our model (*Equations A.46; A.47*) displays two different regimes of growth according to the saturation state of mRNAs. Our fitting procedure is thus divided into two steps. First, we impose an arbitrary transition time $t^{**}$ to determine by a least mean square minimization the three other parameters. Then, we minimize the variance between the obtained solution with the data to determine $t^{**}$. The optimal values of the fitting parameters are

$$t^{**} = 2h44\text{min} \quad , \quad \tau_p = 1h9\text{min} \quad , \quad k_r = 0.62^{-1} \quad , \quad v_1 = 30\text{fL} \tag{A.51}$$

## 5.1 Manning condensation

We give a simple description of the phenomenon of Manning condensation, based on *Barrat and Joanny, 1996*. The electrostatic potential close to an infinitely charged thin rod, in a salt bath, reads

$$\psi = \frac{2 \cdot l_b}{A} \cdot ln(\kappa r) \tag{A.52}$$

where $l_b$ is the Bjerrum length. It is the length at which the electrostatic interaction between two elementary particles is on the order of $k_B T$. Its value in water at room temperature is $l_b \approx 0.7$nm. $A$ is the average distance between two charges on the polymer. $\kappa = \sqrt{8\pi \cdot l_b \cdot (n^+ + n^-)}$ is the inverse of the Debye length. At equilibrium, the distribution of charges around the rod follows a Boltzmann distribution:

$$n^+ = n_0 \cdot e^{-\Psi} = \frac{n_0}{(\kappa r)^{2 \cdot \frac{l_b}{A}}} \tag{A.53}$$

The total number of positive charges per unit length of the rod reads within a distance $\mathcal{R}$:

$$N(\mathcal{R}) = \int_0^{\mathcal{R}} n^+ 2\pi r dr = \frac{2\pi \cdot n_0}{\kappa^{2 \cdot \frac{l_b}{A}}} \cdot \int_0^{\mathcal{R}} \frac{1}{(r)^{2\frac{l_b}{A}-1}} dr \tag{A.54}$$

When $u = \frac{l_b}{A} < 1$, $N(\mathcal{R})$ is dominated by its upper bond and goes to 0 close to the rod. On the other hand, when $u = \frac{l_b}{A} > 1$, $N(\mathcal{R})$ diverges as $\mathcal{R} \to 0$ indicating a strong condensation of the counterions on the rod. This singularity is symptomatic of the breakdown of the linear Debye–Huckel theory. The solution of the nonlinear Poisson–Boltzmann equation shows that there is formation of a tightly bound layer of counterions very near the rod, which effectively decreases the charge density (increases $A$) up to the value $u_{eff} = 1$(*Schiessel, 2014*). It means that if $A$ is smaller than $l_b$ the manning condensation will renormalize $A$ to $A^{eff} = l_b$. The rationale behind this renormalization is to decrease the electrostatic energy of the system by condensing free ions on the polymer. Note that there is an energy penalty associated to the loss of entropy of the condensed counterions. For weakly charged polymers, this loss of entropy is not energetically favorable – case where $u = \frac{l_b}{A} < 1$ and no condensation occurs. If the density of charge of the polymer increases, Manning condensation becomes energetically favorable – case where $u = \frac{l_b}{A} > 1$. By virtue of the high lineic charge of DNA, Manning condensation will be favorable, $u^{DNA} \sim 4$, that is, 75% of the DNA charge is screened by counterion condensation.

## 6.1 The nested Pump-Leak model

The nested Pump-Leak model is described by a set of nonlinear equations, that is, the electroneutrality, the balance of pressures, and the balance of ionic fluxes, in the cytoplasm, subscript $c$, and in the nucleus, subscript $n$. In its most general form, the system reads

$$\begin{cases} n_c^+ - n_c^- - z_p \cdot p_c - z_a \cdot a_c^f = 0 \\ n_n^+ - n_n^- - z_p \cdot p_n - z_a \cdot a_n^f - q = 0 \\ \Delta \Pi_c = \Delta P_c \\ \Delta \Pi_n = \Delta P_n \\ n_c^+ \cdot n_c^- = \alpha_0 \cdot n_0^2 \\ n_n^+ \cdot n_n^- = n_c^+ \cdot n_c^- \\ (n_n^+)^{z_a} \cdot a_n^f = (n_c^+)^{z_a} \cdot a_c^f \end{cases} \tag{A.55}$$

Here, we apply the nested Pump-Leak model to mammalian cells, such that we can neglect the cytoplasmic difference of hydrostatic pressure with respect to the external osmotic pressure. If the nuclear envelope is not under tension, the condition of osmotic balance at the nuclear envelope simply implies that the volume of each compartment takes the same functional form as in the Pump-Leak model:

$$\begin{cases} V_n = R_n + \frac{N_n^{tot}}{2n_0} \\ V_c = R_c + \frac{N_c^{tot}}{2n_0} \end{cases} \tag{A.56}$$

It is thus straightforward to show that both the volume of the nucleus and the volume of the cytoplasm scale with each other (*Equation 12*).

### 6.1.1 Dry volumes in the nucleus and in the cytoplasm

We assume that the dry volumes in the nucleus and in the cytoplasm are proportional to the total volumes of each compartments and are equal to each other: $R_n = r \cdot V_n$ and $R_c = r \cdot V_c$. Under this assumption, the NC ratio simply becomes the ratio of the wet volumes:

$$NC = \frac{V_n}{V_c} = \frac{V_n - R_n}{V_c - R_c} \tag{A.57}$$

This hypothesis is practical rather than purely rigorous. It is based on experiments that suggest that dry mass occupies about 30% of the volume of both the nucleus and the cytoplasm for several cell types and conditions (*Venkova et al., 2022*; *Lemière et al., 2022*; *Rowat et al., 2006*). Nonetheless, even if this assumption were to be inexact, our discussion would then rigorously describe the slope of the linear relationship between nucleus and cell volume (*Equation 12*) which was shown to be robust to perturbation (*Guo et al., 2017*).

### 6.1.2 Membrane potential in the simple Pump-Leak model

We insert the expression for the volume (*Equation A.14*) and ion concentrations (*Equation A.10*) into the expression of the transmembrane potential found in the simple Pump-Leak model (*Equation A.8*) to show that a nonvanishing potential exists as soon as there are trapped charged particles. In the simple Pump-Leak model, the plasma membrane potential difference hence reads

$$\begin{cases} U = ln\left(\frac{-z \cdot (-1+r) + \sqrt{z^4 + \alpha_0 - z^2 \cdot (-1 + \alpha_0 + 2 \cdot r)}}{z^2 - 1}\right) \\ \text{With,} \quad r = \sqrt{1 + (z^2 - 1) \cdot (1 - \alpha_0)} \end{cases} \tag{A.58}$$

We find that $U$ monotonically increases (in absolute value) with the average charge of the cell trapped components. This differs from the nuclear membrane potential that vanishes when the charge of the chromatin is diluted regardless of the properties of the trapped proteins (*Equation 16*).

### 6.1.3 General formula for the regime $NC_2$, that is, no metabolites

As stated in the main text, an important limit regime, $NC_2$, is achieved when there are no metabolites in the cell. Specifically, the previous system of equations becomes uncoupled with respect to the nuclear and cytoplasmic set of variables such that we can solve the system analytically. Using the exact same algebra as used in the simple Pump-Leak model, we express the volumes and the NC ratio as

$$
\begin{cases}
V_{tot} = (R_c + R_n) + \dfrac{N_n^{tot} + N_c^{tot}}{2n_0} \\[2mm]
N_c^{tot} = P_c \cdot \dfrac{z_p^2 - 1}{-1 + \sqrt{1 + (1 - \alpha_0) \cdot (z_p^2 - 1)}} \\[2mm]
N_n^{tot} = P_n \cdot \dfrac{(z_{n,eff}^2 - 1)}{-1 + \sqrt{1 + (1 - \alpha_0)(z_{n,eff}^2 - 1)}} \\[2mm]
z_{n,eff} = z_p + \dfrac{Q^{eff}}{P_n} \\[2mm]
NC = NC_2 = NC_1 \cdot \dfrac{(z_{n,eff}^2 - 1)}{(z_p^2 - 1)} \cdot \dfrac{-1 + \sqrt{1 + (1 - \alpha_0)(z_p^2 - 1)}}{-1 + \sqrt{1 + (1 - \alpha_0)(z_{n,eff}^2 - 1)}}
\end{cases}
\tag{A.59}
$$

### 6.1.4 Analytical solutions in the regime $z_p = 1$, $z_a = 1$, and $\alpha_0 \sim 0$

In this regime of high pumping, no anions occupy the cell. We simplify the notations by denoting by $n$ the concentration of cations. The system of equation to solve is stated in the main text (**Equation 15**). We first express the concentrations of cations and metabolites in the cytoplasm and nucleus as a function of $n_0$, $q^{eff}$, and $p$ thanks to the electroneutrality equations and balance of osmotic pressures:

$$
\begin{cases}
n_n = n_0 + \dfrac{q^{eff}}{2} \\[2mm]
d_n^f = \left( n_0 - \dfrac{q^{eff}}{2} \right) - p_n \\[2mm]
n_c = n_0 \\[2mm]
d_c^f = n_0 - p_c
\end{cases}
\tag{A.60}
$$

This allows us to write the nuclear envelope potential as **Equation 16** in the main text. Using the balance of nuclear osmotic pressure, we express the nuclear volume function of the number of nuclear osmolytes:

$$
V_n - R_n = \frac{Q^{eff} + 2A_n^f + 2P_n}{2n_0}
\tag{A.61}
$$

This implies that the nuclear envelope potential can be written without the dependence on $n_0$ as in **Equation 16**. We then express the NC ratio in two different manners. First, using the interpretation of wet volumes, namely, the total number of osmolytes in the compartments over $2n_0$. Second, we take advantage of the concentrations of metabolites and cations in **Equation A.60** to express the ratio of protein concentrations. After simple algebra, we obtain:

$$
\begin{cases}
NC = \dfrac{P_n}{P_c} \cdot \dfrac{p_c}{p_n} = NC_1 \cdot \left( 1 + \dfrac{Q^{eff}}{2P_n + 2A_n^f + Q^{eff}} + \dfrac{Q^{eff^2}}{2P_n + 2A_n^f + Q^{eff}} \right) \\[2mm]
NC = \dfrac{1}{2} \cdot \dfrac{2A_n^f + 2P_n + Q^{eff}}{A_{tot}^f - A_n^f + P_c}
\end{cases}
\tag{A.62}
$$

For clarity, we now normalize each number by $2P_n$, for example, $\overline{A}_{tot} = \frac{A_{tot}}{2P_n}$. Equating both expressions of the NC ratio leads to a second order polynomial in $\overline{A}_n^f$:

$$
2\left(1 + \tfrac{1}{NC_1}\right) \cdot \left(\overline{A}_n^f\right)^2 + \left(-2\overline{A}_{tot} + (1 + \overline{Q}^{eff})^2 + \tfrac{1}{NC_1} \cdot (1 + 2\overline{Q}^{eff})\right) \cdot \overline{A}_n^f - \overline{A}_{tot} \cdot (1 + \overline{Q}^{eff})^2 = 0
\tag{A.63}
$$

The solution $\overline{A}_n^f$ now reads

$$
\overline{A}_n^f = \frac{2\overline{A}_{tot} - \tfrac{1}{NC_1} \cdot (1 + 2\overline{Q}^{eff}) - (1 + \overline{Q}^{eff})^2 + \sqrt{\left(2\overline{A}_{tot} - \tfrac{1}{NC_1} \cdot (1 + 2\overline{Q}^{eff}) - (1 + \overline{Q}^{eff})^2\right)^2 + 8 \cdot (1 + \tfrac{1}{NC_1}) \cdot \overline{A}_{tot} \cdot (1 + \overline{Q}^{eff})^2}}{4 \cdot (1 + \tfrac{1}{NC_1})}
\tag{A.64}
$$

which leads to the following expression for NC:

$$
NC = NC_1 \cdot \frac{2\overline{A}_{tot} + \tfrac{1}{NC_1} + (1 + \overline{Q}^{eff})^2 + \sqrt{\left(2\overline{A}_{tot} - \tfrac{1}{NC_1} \cdot (1 + 2\overline{Q}^{eff}) - (1 + \overline{Q}^{eff})^2\right)^2 + 8 \cdot (1 + \tfrac{1}{NC_1}) \cdot \overline{A}_{tot} \cdot (1 + \overline{Q}^{eff})^2}}{2 \cdot \left(1 + 2\overline{A}_{tot} + \tfrac{1}{NC_1} + \overline{Q}^{eff}\right)}
\tag{A.65}
$$

As a sanity check, we verify some asymptotic expressions discussed in the main text. For example, when $\overline{Q}^{eff} \ll 1$ or $\overline{A}_{tot} \gg 1$, we recover that NC becomes equal to $NC_1$. On the other hand, when $\overline{A}_{tot} \ll 1$, we recover that $NC = NC_1 \cdot (1 + \overline{Q}^{eff}) = NC_2$

### 6.1.5 Control parameters of the nested Pump-Leak model during growth

The precise value of the parameter $NC_1 = \frac{P_n}{P_c}$ is biologically set by an ensemble of complex active processes ranging from transcription, translation to the Ran GTPase cycle, and nuclear transport. The precise modeling of nucleo-cytoplasmic transport is out of the scope of this article but could easily be incorporated to our framework. Nonetheless, we can safely assume that nucleo-cytoplasmic transport is fast compared to the typical timescale of growth. In this case, neglecting protein degradation on the timescale of the G1 phase, the total number of proteins in the nucleus is simply the number of proteins assembled that possessed a nuclear import signal (NIS) in their sequence. Using the same notation as earlier, in the exponential growth regime, the total number of proteins in the nucleus reads

$$P_{tot,n}(t) = \sum_{j \in NIS} \frac{\phi_j}{\phi_r} \cdot P_r(t) \tag{A.66}$$

where $P_r(t)$ accounts for the number of ribosomes, $\phi_j$ is the fraction of genes coding for the protein $j$ (see Appendix 1, section 4.1). The subscript $j$ is summed over the genes coding for proteins having nuclear import signals in their sequence. Proteins in the nucleus can either be DNA bound or unbound. For example, histones or DNA polymerases bind to the DNA. Only the unbound proteins contribute to the osmotic pressure. Denoting $k_{u,j}$ and $k_{b,j}$ the reaction rate of binding and unbinding of protein $j$ and assuming that the reactions of binding and unbinding are fast compared to the timescale of growth, we finally express the number of free proteins in the nucleus as

$$P_{free,n}(t) = \sum_{j \in NIS} \frac{k_{u,j}}{k_{b,j} + k_{u,j}} \cdot \frac{\phi_j}{\phi_r} \cdot P_r(t) \tag{A.67}$$

It is then straightforward to express $NC_1$ as

$$NC_1 = \frac{\sum_{j \in NIS} \frac{k_{u,j}}{k_{b,j} + k_{u,j}} \cdot \frac{\phi_j}{\phi_r}}{\sum_{j \notin NIS} \frac{k_{u,j}}{k_{b,j} + k_{u,j}} \cdot \frac{\phi_j}{\phi_r}} \tag{A.68}$$

An important result of this abstract modeling is that $NC_1$ is independent of time during the exponential growth due to the fact that both $P_n(t)$ and $P_c(t)$ are proportional to $P_r(t)$, which is why we adopted it as a control parameter. The same goes for our second control parameter $\frac{A_{tot}}{2P_n}$, which is also constant during exponential growth.

### 6.1.6 Phase diagram

In this section, we address the case $\Delta P_n \neq 0$ and assume that the cell does not adhere to the substrate such that we consider the nucleus to be spherical. For simplicity, we neglect the dry volume because we want to consider hypo-osmotic shock experiment where dry mass will be diluted, making a dry volume a second order effect of the order 10%. We first make the problem dimensionless. There are two dimensions in our problem: an energy and a length. This means that we can express all our parameters that possess a dimension with a unit energy and a unit length. Moreover, we have three parameters with physical dimensions: the extracellular osmolarity $n_0$, the nuclear envelope tension $\gamma$, and the thermal energy $k_B T$. The theorem of **Buckingham, 1914** tells us that we can fully describe our problem with a single dimensionless parameter and the three parameters with by definition no dimensions. In this geometry, we choose $[\left(\frac{4\pi}{3}\right)^{1/3} \cdot \frac{\gamma_0}{k_B T X_n^{1/3} n_0^{2/3}}, \alpha_0, z_n, X_n]$. Laplace law reads:

$$\Delta P_n = \frac{2\gamma_n}{(\frac{3}{4\pi} V_n)^{1/3}} \tag{A.69}$$

Using the following dimensionless quantities:

$$\begin{cases} \overline{V}_n = \frac{2n_0}{X_n} \cdot V_n \\ \overline{\gamma}_n = \left(\frac{4\pi}{3}\right)^{1/3} \cdot \frac{\gamma_0}{k_B T X_n^{1/3} n_0^{2/3}} \end{cases} \tag{A.70}$$

Equality of pressures becomes

$$\sqrt{\left(\frac{z_n}{\overline{V}_n}\right)^2 + \alpha_0} + \frac{1}{\overline{V}_n} - 1 - \frac{\overline{\gamma}_n}{\overline{V}_n^{1/3}} = 0 \tag{A.71}$$

*Equation A.71* cannot be solved analytically for $\overline{V}_n$. However, five asymptotic regimes can be identified (see *Appendix 1—figure 1*):

$$\begin{cases} V_1 = \left(\frac{3}{16\pi}\right)^{\frac{1}{2}} \cdot \left(\frac{k_B T X_n}{z_n \cdot \gamma_n}\right)^{\frac{3}{2}} \\ V_2 = \left(\frac{3}{2^{10} \cdot \pi}\right)^{\frac{1}{5}} \cdot \left(\frac{k_B T \cdot z_n^2 \cdot X_n^2}{n_0 \cdot \sqrt{\alpha_0} \cdot \gamma_n}\right)^{\frac{3}{5}} \\ V_3 = \left(\frac{3}{16\pi}\right)^{\frac{1}{2}} \cdot \left(\frac{k_B T \cdot X_n}{\gamma_n}\right)^{\frac{3}{2}} \\ V_4 = \frac{X_n}{2 n_0} \cdot \frac{1}{1 - \sqrt{\alpha_0}} \\ V_5 = \frac{X_n}{2 n_0} \cdot \frac{z_n}{\sqrt{1 - \alpha_0}} \end{cases} \tag{A.72}$$

- $V_4$ and $V_5$ are the limit regimes where osmotic pressure is balanced at the nuclear envelope.
- $V_3$ is the limit regime where the difference of osmotic pressure is dominated by the impermeant molecules trapped inside the nucleus. This happens when the proteins are not or very weakly charged. This difference of osmotic pressure is balanced by the Laplace pressure of the lamina.
- $V_1$ is the limit regime where the difference of osmotic pressure is dominated by the counterions of the impermeant molecules. This difference of osmotic pressure is balanced by the Laplace pressure of the lamina.
- $V_2$ is an intermediate regime that can arise when $\alpha_0 \approx 1$. The difference of osmotic pressure takes the form of $\Delta\Pi_n \approx k_B T \cdot \frac{1}{\sqrt{\alpha_0}} \cdot \frac{(z_n \cdot x_n)^2}{4 n_0}$. This osmotic pressure defines an effective virial coefficient between monomers of DNA and proteins $v_{el} = \frac{1}{\sqrt{\alpha_0}} \frac{z_n^2}{2 n_0}$. This difference of osmotic pressure is balanced by the Laplace pressure at the nuclear envelope.
- Note that when $\alpha_0 \approx 0$ (strong pumping), only the counterion necessary for electroneutrality remain in the nucleus. $\Pi_n$ is simply $(z_n + 1) \cdot x_n$ and is either balanced by the Laplace pressure of the lamina or the external osmotic pressure (see *Appendix 1—figure 1*).

Finally, the analytical expressions for the crossover lines $\overline{\gamma}_{i,j}$ between regime of volume $V_i$ and volume $V_j$, plotted in *Appendix 1—figure 1* read

$$\begin{cases} \overline{\gamma}_{1,2} = (4 \cdot \alpha_0 \cdot z_n)^{\frac{1}{3}} \\ \overline{\gamma}_{1,4} = z_n \cdot (1 - \sqrt{\alpha_0})^{\frac{2}{3}} \\ \overline{\gamma}_{1,5} = z_n^{\frac{1}{3}} \cdot (1 - \alpha_0)^{\frac{1}{3}} \\ \overline{\gamma}_{2,5} = \frac{z_n^{\frac{1}{3}} \cdot (1 - \alpha_0)^{\frac{5}{6}}}{2 \cdot \sqrt{\alpha_0}} \\ \overline{\gamma}_{2,4} = \frac{z_n^2 \cdot (1 - \sqrt{\alpha_0})^{\frac{5}{3}}}{2 \cdot \sqrt{\alpha_0}} \\ \overline{\gamma}_{2,3} = z_n^{-\frac{4}{3}} \cdot (4 \cdot \alpha_0)^{\frac{1}{3}} \\ \overline{\gamma}_{3,5} = z_n^{-\frac{2}{3}} \cdot (1 - \alpha_0)^{\frac{1}{3}} \\ \overline{\gamma}_{3,4} = (1 - \sqrt{\alpha_0})^{\frac{2}{3}} \\ z_{1,3} = 1 \\ z_{4,5} = \frac{\sqrt{1 - \alpha_0}}{1 - \sqrt{\alpha_0}} \end{cases} \tag{A.73}$$

### 6.1.7 Saturating volume after a hypo-osmotic shock

The saturation occurs when the nuclear osmotic pressure is balanced by the Laplace pressure making nuclear volume insensitive to the external osmolarity (*Equation A.72*). We assume that the nuclear envelope behaves elastically with a stretching modulus $K$ beyond a surface area $S^*$ for which nuclear envelope folds are flattened:

$$\gamma_n = \begin{cases} 0 & , \text{ if } S_n \leq S^* \\ K \cdot \left(\frac{S_n}{S^*} - 1\right) & , \text{ if } S_n \geq S^* \end{cases} \tag{A.74}$$

As justified in the main text, metabolites tend to leave the nucleus with decreasing external osmolarity. The saturating volume is obtained when $\Delta P_n \gg \Pi_0$ and $A_n^f \ll P_n$. From *Equation A.15* applied to the volume of the nucleus, we thus obtain

$$\Delta P = (z_n^{eff} + 1) \cdot \frac{P_n}{V_n^{max}} \tag{A.75}$$

where $z_n^{eff} = z_p + \frac{Q^{eff}}{P_n}$. Similarly to the last subsection, we normalize tensions by $(\frac{3}{4\pi})^{1/3} \cdot k_B T \cdot P_n^{1/3} \cdot n_0^{2/3}$ and volumes by $\frac{2 \cdot P_n}{2n_0}$. *Equation A.75* leads to the equation ruling the saturating volume:

$$(v_n^{max})^{4/3} - (v_n^{iso})^{2/3} \cdot (1+s) \cdot (\frac{n_0}{n_0^{iso}})^{2/3} \cdot (v_n^{max})^{2/3} - (v_n^{iso})^{2/3} \cdot (1+s) \cdot (\frac{n_0}{n_0^{iso}})^{2/3} \cdot \frac{\frac{1}{2} \cdot (z_n^{eff}+1)}{\overline{K}} = 0 \tag{A.76}$$

where $s = \frac{S^*}{S_n^{iso}} - 1$ is the fraction of folds that the nucleus possesses at the isotonic osmolarity. $v_n^{max} = \frac{2n_0 \cdot V_n^{max}}{2P_n}$ is the normalized saturating nuclear volume and $v_n^{iso}$ the normalized nuclear volume at the isotonic osmolarity. $\overline{K} = \frac{K}{(\frac{3}{4\pi})^{1/3} \cdot k_B T \cdot P_n^{1/3} \cdot n_0^{2/3}}$ is the normalized effective stretching modulus of the nuclear envelope. Solving the previous equation, coming back to real volumes $\frac{V_n^{max}}{V_n^{iso}} = \frac{v_n^{max}}{v_n^{iso}} \cdot \frac{n_0^{iso}}{n_0}$ and Taylor develop the result for $n_0 \longrightarrow 0$ leads to *Equation 19* in the main text.

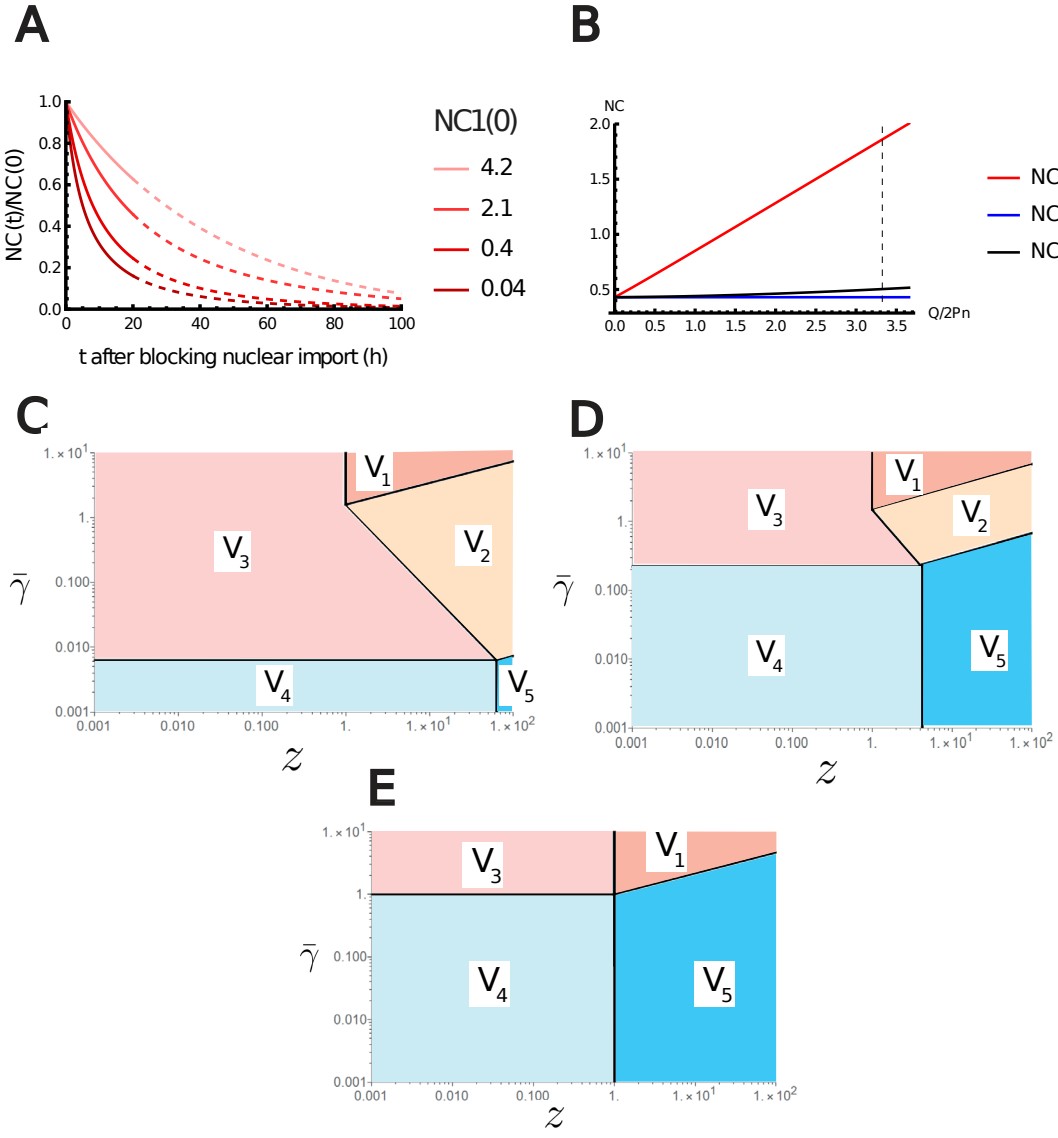

**Appendix 1—figure 1.** Additional results of the nested Pump-Leak model. (**A**) Variation of the nuclear-to-cytoplasmic (NC) ratio during growth after blocking nuclear import. (**B**) Variations of the NC ratio according to the effective charge of the chromatin normalized by the number of trapped proteins in the nucleus $\frac{Q}{2P_n}$. The NC ratio is bounded by two limit regimes. NC1, if the number of metabolites is assumed infinite. NC2, if there are no metabolites. The vertical black dashed line depicts the value of $\frac{Q}{2P_n}$ estimated in Appendix 1, section 3.1 for diploid mammalian cells. (**C–E**) log–log plot of the different regimes of V **Equation A.72** in the plan $(z_n, \overline{\gamma}_n)$ for $\alpha_0$ fixed (**C**) $\alpha_0 = 0.99$, (**D**) $\alpha_0 = 0.8$, and (**E**) $\alpha_0 = 0.001$. The crossover lines plotted are given in **Equation A.73**.

## 6.1.8 Geometrical impact

The previous equations were conducted for a spherical geometry. Interestingly, while the precise geometry does not qualitatively change our results, we expect the saturation of nuclear volume to occur more easily for a pancake shape – a shape closer to the shape of adhered cells. Indeed, the scaling between surface and volume is approximately linear in this case: $V \sim h \cdot S$, while it is sublinear for spheres $S \sim V^{2/3}$. Thus, smaller osmotic shocks will be required to tense the nuclear envelope and so as to reach the saturating regime.

## 7.1 Electrostatic interactions are encompassed within our framework

We directly compute the contribution of electrostatic interactions to the osmotic pressure based on *Barrat and Joanny, 1996*. The total interaction energy of a solution of charged particles of average density $x$ within a volume $V$ is, using the Poisson–Boltzmann framework,

$$\frac{E_{el}}{k_B T} = \frac{l_b \cdot z^2}{2} \int \int x(\vec{r}) \cdot x(\vec{r'}) \cdot \frac{e^{-\kappa|\vec{r}-\vec{r'}|}}{|\vec{r}-\vec{r'}|} \quad d^3\vec{r} \cdot d^3\vec{r'} \tag{A.77}$$

where $x(\vec{r})$ is the local density of impermeant molecules in the cell. Fourier analysis allows us to rewrite this equation:

$$\frac{E_{el}}{k_B T} = \frac{l_b \cdot z^2}{2} \int x(\vec{k}) \cdot x(-\vec{k}) \cdot \frac{4\pi}{k^2+\kappa^2} \quad d^3\vec{k} \approx \frac{l_b \cdot z^2}{2} \cdot x^2 \cdot \frac{4\pi}{\kappa^2} \cdot V \tag{A.78}$$

From which, we derive the expression of the osmotic pressure:

$$\frac{\Pi_{el}}{k_B T} \approx \frac{1}{2} \cdot \frac{z^2}{2n_0} \cdot x^2 \tag{A.79}$$

We now show that this term is already encompassed within our framework. For the simplicity of the discussion, we neglect pumping, that is, $\alpha_0 \sim 1$. The difference of osmotic pressure then reads (see *Equation A.11*)

$$\frac{\Delta\Pi}{k_B T} = \sqrt{(zx)^2 + 4n_0^2} + x - 2n_0 \tag{A.80}$$

which, under the right regime, that is, $zx << 2n_0$, leads to the same term. As mentioned above, this osmotic pressure defines an effective electrostatic virial coefficient between monomers:

$$v_{el} = \frac{z^2}{2n_0} \tag{A.81}$$

## 8.1 Possible extension to explain the scaling of other organelles

Organelles are also known to display characteristic scaling trends with cell size (*Chan and Marshall, 2010*). We emphasize that these scalings may be of different origins and would require a much more careful treatment with respect to the specificity of the organelle. Nevertheless, in this subsection, we highlight that, if we assume that the organelles are constrained by the same physical laws, our model then easily extends to the inclusion of other organelles and in particular would predict the proportionality between organelle and cell volumes.

We model an organelle in our theory by a compartment bound by a membrane that trap some molecules. For the sake of generality, we assume that there is an active transport of cations through this membrane. As a matter of coherence with the previous notations, we will call by $\alpha_{org} = e^{-\frac{P_{org}}{k_B T g+}}$ the parameter that compares the active pumping through the organelle's membrane versus the passive leakage. Donnan equilibrium on both side of the organelle reads

$$n_{org}^+ \cdot n_{org}^- = \alpha_{org} \cdot (n_c^+ \cdot n_c^-) = \alpha_{org} \cdot \alpha_0 \cdot n_0^2 \tag{A.82}$$

Hence, the results derived previously also apply to the organelle provided the parameter $\alpha_0$ is changed into $\alpha_{org} \cdot \alpha_0$. Interestingly, in the case of osmotic balance at the membrane of the organelle, it is straightforward to show that the volume of the organelle also scales with the cell volume:

$$\begin{cases} V_{org} = \left(\frac{N_{org}^{tot}}{N^{tot}}\right) \cdot V_{tot} + \left[\left(\frac{N_c^{tot}+N_n^{tot}}{N^{tot}}\right) \cdot R_{org} - \left(\frac{N_{org}^{tot}}{N^{tot}}\right) \cdot (R_c + R_n)\right] \\[2em] N^{tot} = N_c^{tot} + N_n^{tot} + N_{org}^{tot} \\[2em] N_{org}^{tot} = X_{org} \cdot \frac{(z_{org}^2-1)}{-1+\sqrt{1+(1-\alpha_0 \cdot \alpha_{org})(z_{org}^2-1)}} \end{cases} \tag{A.83}$$

