## [Editor Report]

This important theoretical work deals with the problem of homeostasis of protein density within cells, relying on the Pump and Leak model. The model makes predictions both for growing and senescent cells, which they convincingly compare to experimental data on budding yeast. The authors further study the long-standing problem of nuclear scaling (the constant ratio between the nucleus and cell volume), and show that within their model it arises naturally from osmotic balance at the nuclear envelope, with metabolites playing a major role.

---

## [Decision Letter]

**Decision letter after peer review:**

Thank you for submitting your article "Cell size scaling laws: a unified theory" for consideration by *eLife*. Your article has been reviewed by 2 peer reviewers, and the evaluation has been overseen by a Reviewing Editor and Naama Barkai as the Senior Editor. The following individual involved in the review of your submission has agreed to reveal their identity: Jie Lin (Reviewer #1).

Essential revisions:

*Reviewer #2 (Recommendations for the authors):*

1. Are there exceptions to cell size control following the framework presented here? For example, what about glycerol synthesis activated by yeast cells to regulate their volume rather than ion pumps?

2. The authors state "Cells have two main ways to counteract the osmotic imbalance. They can adapt to sustain a high hydrostatic pressure difference as plants do by building cellulose walls. Or, as done by mammalian cells, they can use ion pumps to actively decrease the number of ions inside the cells, thus decreasing the osmotic pressure difference across the cell membrane and therefore impeding water penetration."

However, there is at least one other way, as some cells use water pumps, not ion pumps or the cell wall. Water can be transported across cells in the absence of a driving force; aquaporin channels for example as mentioned by two of the authors in this preprint about cell volume regulation: https://www.biorxiv.org/content/10.1101/2022.08.24.505072v1.full.pdf.

3. The authors state "Our results suggest that the fact that cell division occurs in the exponential regime is essential to prevent cells from being diluted." In theory, it makes sense that exponential growth leads to a time-independent NC ratio. However, is there a significant quantitative difference in practice?

4. For the comment "Note that only half of the proteins are trapped in the nucleus because about half of them have a mass smaller than the critical value 30-60kDa [24]", can the authors supply additional reference(s) for the fraction of 1/2? It seems too high.

5. Some references appear to be missing or incorrect. For example, the authors state "as shown by genetic screen experiments done on fission yeast mutants [3]" but it appears they may have meant to cite this paper(s) from the same lab instead: "A systematic genetic screen identifies essential factors involved in nuclear size control" or possibly this one "A systematic genomic screen implicates nucleocytoplasmic transport and membrane growth in nuclear size control", since these are the papers that actually looked at a genetic screen in *S. pombe*. Ref. [3] is a review paper.

6. Authors should consider discussing their predictions in light of data in Odermatt et al., "Variations of intracellular density during the cell cycle arise from tip-growth regulation in fission yeast" (2021). In particular, the statements "Our findings illustrate a general mechanism by which density is regulated through controlling the relative rates of volume growth and biosynthesis" and "These measurements suggest that the density increase during mitosis and cytokinesis is a consequence of continued mass accumulation when volume growth halts" seem highly related. Similarly for Knapp et al., "Decoupling of Rates of Protein Synthesis from Cell Expansion Leads to Supergrowth" (2019).

---

## [Author Response]

Reviewer #2 (Recommendations for the authors):1. Are there exceptions to cell size control following the framework presented here? For example, what about glycerol synthesis activated by yeast cells to regulate their volume rather than ion pumps?

This comment was partially addressed in the answer of the major comment 1 of the reviewer 2. We copy some part of the latter answer below for convenience. We emphasize that while the laws and coarse grained parameters are general. The limited set of microscopic phenomena described such as the counterion release for the mitotic overshoot, the saturation of DNA and mRNA for the dry mass dilution, and the nuclear envelope stretching for the breakdown of the nuclear scaling upon osmotic shock experiment, is not. Thus, there are a number of other biological phenomena that affects cell size. For example, again based on a Pump-Leak model, recent studies [Venkova, *eLife*, 2022] have shown that the change of mechanical tension at the plasma membrane during cell spreading alters the permeability of ion channels, resulting in a volume adaptation. As stated by the reviewer, another important way cells have to control their size is through the synthesis of osmolytes such as glycerol for budding yeast cells upon hypertonic stress. In our theory it is very well possible to consider the case where there is a cell wall pressure difference ∆*P* and no ion pumps: *α*_0_ = 1. Moreover, glycerol is included inside the category of metabolites. If, because of the activation or inactivation of various signalling pathway such as mTORc or a Map Kinase pathway [Neurohr, Trends in Cell Biology], the number of glycerol molecules would increase in the cell, then the volume in our theory would also increase because *X* would increase.

To take into account the reviewer point we modify the text:

We add in the introduction: “On the other hand, the absence of linear scaling relation between protein and small osmolyte numbers is at the root of the breakdown of density homeostasis. This conclusion is in line with two biological mechanisms that were proposed to explain the regulation of size and density respectively during cell spreading and under hyper-osmotic shocks. Recent studies [Venkova, *eLife*, 2022], [Adar, PNAS, 2020] have indeed shown that the change of mechanical tension at the plasma membrane during cell spreading alters the permeability of ion channels, and results in a volume adaptation at constant dry mass. Another important way through which cells control their size independently of their mass is through metabolite synthesis. An example is the synthesis of glycerol that occurs in budding yeast following a hypertonic shock [Neurohr, 2020]. While these two mechanisms are now well established, here, we propose a new physical interpretation of two other important biological events, namely the dilution that occurs at senescence and the dilution that occurs at the beginning of mitosis.”

We add in the discussion: “We emphasize that while we claim that the physical laws and the coarse-grained physical parameters that constrain cell size are ubiquitous, the specific set of biological processes described in our paper at the root of the variation of such parameters, is not. In particular, the new biological mechanism, namely the saturation of DNA and mRNA, proposed here to explain dry mass dilution at senescence, is not the only biological mechanism that affects dry mass density. We can indeed quote at least two other identified processes. Variations of the mechanical tension at the plasma membrane during cell spreading was shown to alter the permeability of ion channels, resulting in a volume adaptation at constant dry mass ([Venkova, 2022],[Adar, 2020]). Similarly, activation of metabolic pathway synthesis such as glycerol for budding yeast cells upon hypertonic stress [Neurohr, 2020], allows to recover cell volume likely at almost constant dry mass (since glycerol is a metabolite). Nevertheless the latter two processes can be easily including in our framework.”

2. The authors state "Cells have two main ways to counteract the osmotic imbalance. They can adapt to sustain a high hydrostatic pressure difference as plants do by building cellulose walls. Or, as done by mammalian cells, they can use ion pumps to actively decrease the number of ions inside the cells, thus decreasing the osmotic pressure difference across the cell membrane and therefore impeding water penetration."However, there is at least one other way, as some cells use water pumps, not ion pumps or the cell wall. Water can be transported across cells in the absence of a driving force; aquaporin channels for example as mentioned by two of the authors in this preprint about cell volume regulation: https://www.biorxiv.org/content/10.1101/2022.08.24.505072v1.full.pdf.

To our knowledge there is no identified molecular water pumps and the paper quoted does not state that. Effective water pumps can nevertheless be mentioned such as contractile vacuoles in paramecia [Tani, Cell Biology International, 2002]. We are however not aware of similar effective water pumps in yeasts, mammalian cells or bacteria.

3. The authors state "Our results suggest that the fact that cell division occurs in the exponential regime is essential to prevent cells from being diluted." In theory, it makes sense that exponential growth leads to a time-independent NC ratio. However, is there a significant quantitative difference in practice?

We do not know how to quantitatively test this question. What was shown however, is that diluted cells have cell cycle defects. This may be a determinant of stem cell potential during ageing [Lengefeld, Science advances, 2021] or may cause a functional decline towards senescence in fibroblasts [Neurohr, Cell, 2019]. While this argument is qualitative, we still think that it is worth mentioning that cells that would remain in the non-exponential regime of growth for too long will be also diluted and are thus likely to have their function impacted.

To take into account reviewer’s point we modify our sentence:

“Our results suggest that both regimes of growth are not equivalent for proper cell function. Cells that would remain in the "non-exponential" growth regime for too long will be diluted. This may be at the root of the observed cell cycle defects [Neurohr, Cell, 2019] and be the cause of functional decline towards senescence in particular for fibroblasts [Neurohr, Cell, 2019] and hematopoietic stem cells [Lengefeld, Science advances, 2021]. We think that a more precise understanding of the relationship between dilution and cell cycle defects remains an exciting avenue for future research.”

4. For the comment "Note that only half of the proteins are trapped in the nucleus because about half of them have a mass smaller than the critical value 30-60kDa [24]", can the authors supply additional reference(s) for the fraction of 1/2? It seems too high.

The exact reference we are referring to is: http://book.bionumbers.org/how-big-is-the-average-protein/ If we consider that the average length of a mammalian cell protein is 400 a.a and that the average mass of an amino-acid is 110 Da, then the average mass of a protein is 44 kDa which is roughly the critical mass for passage through a nuclear pore. We acknowledge that the average may be different from the median of the distribution and that we might be overestimated the claimed 1/2 fraction. Nevertheless, for the argument we make here, we do not need the precise value of 1/2 since we just wanted to show that even if the proteins were not all trapped in the nucleus, this would not impinge on our conclusions. We thus decide to modify the sentence into:

“Note that proteins that have a mass smaller than the critical value 30-60kDa are not trapped in the nucleus as they can freely diffuse throughout the nuclear pores. This nevertheless represents more a semantic issue than a physical one …”.

5. Some references appear to be missing or incorrect. For example, the authors state "as shown by genetic screen experiments done on fission yeast mutants [3]" but it appears they may have meant to cite this paper(s) from the same lab instead: "A systematic genetic screen identifies essential factors involved in nuclear size control" or possibly this one "A systematic genomic screen implicates nucleocytoplasmic transport and membrane growth in nuclear size control", since these are the papers that actually looked at a genetic screen in *S. pombe*. Ref. [3] is a review paper.

We thank the reviewer for highlighting this imprecision. The exact reference that we now quote in the manuscript is:

“Cantwell H, Nurse P (2019) A systematic genetic screen identifies essential factors involved in nuclear size control. PLoS Genet”

6. Authors should consider discussing their predictions in light of data in Odermatt et al., "Variations of intracellular density during the cell cycle arise from tip-growth regulation in fission yeast" (2021). In particular, the statements "Our findings illustrate a general mechanism by which density is regulated through controlling the relative rates of volume growth and biosynthesis" and "These measurements suggest that the density increase during mitosis and cytokinesis is a consequence of continued mass accumulation when volume growth halts" seem highly related. Similarly for Knapp et al., "Decoupling of Rates of Protein Synthesis from Cell Expansion Leads to Supergrowth" (2019).

We thank the reviewer for letting us know about this literature. While our theory may indeed be able to expand the theoretical understanding of these observations, we let the precise comparison between the suggested data and our model for future studies. We nevertheless quote these references in our manuscript because they are indeed highly related to our study.